# String amplitudes and mutual information in confining backgrounds: the partonic behavior

Mahdis Ghodrati[a],

[a]*Asia Pacific Center for Theoretical Physics, Pohang University of Science and Technology, Pohang 37673, Republic of Korea*

*e-mails: mahdis.ghodrati@apctp.org*

## Abstract

In the background of several holographic confining backgrounds, we present the connections between the behaviors of string scattering amplitudes and mutual information. We lay down the analogies between the logarithmic branch cut behavior of the string scattering amplitude in $4d$, $\mathcal{A}_4$, at low and medium Mandelstam variable $s$ observed in [1], which is due to the dependence of the string tension on the holographic coordinate, and the branch cut behavior observed in mutual information and critical distance $D_c$ at low-cut-off variable $u_{KK}$ studied in our previous work [2]. It can also be seen that in both cases, as $s$ or $u_{KK}$ increases, the peaks in the branch cuts fade away in the form of $\text{Re}[\mathcal{A}_4] \propto s^{-1}$. Then, we used modular flow and modular Hamiltonian as intermediary concept to further clarify the observed connection. We discussed how mutual information itself can detect chaos in various scenarios. In addition, we considered two examples of Compton scattering between two photons and also the decay of a highly excited string into two tachyons and scrutinized the pattern of entanglement entropy and the change in the mutual information in these examples. Then, the kink-kink and kink-antikink scatterings as simple models of scattering in confining geometries have been used to examine the fractal structures in the scatterings of topological defects. Finally, the relationships between Regge conformal block and quantum error correction codes through pole-skipping and chaos bound have been postulated. These observed connections can further establish the ER=EPR conjecture and the general interdependence between the scattering amplitudes and entanglement entropy.

# 1   Introduction

In the SLAC laboratory, it has been observed that at high energy and fixed angle, the hadronic scattering amplitudes fall off based on a power low behavior in the Mandelstam variable $s$ (the center of mass energy), and not based on an exponential behavior in $s$, as has been previous to this experiment people thought. This observation is in fact incompatible with the results coming from the string theory. On the other hand, using holography, an interesting connection between entanglement entropy (EE) and the string scattering amplitude has been noticed in [3–6]. Additionally, the correspondence between strings structures and black hole pointed to the fact that one could get a lot of information from the structures of string amplitude as well.

According to these connections [3–15], we can now show that the richer mixed correlation measures such as negativity and mutual information (MI) can further establish connections between the behaviors of scattering amplitudes and the pattern of quantum entanglement. In [16], the scattering amplitude of highly excited strings has been studied within the DDF (Del Giudice, Di Vecchia, Fubini [17]) formalism where it has been implemented such that the excited string is produced by photons that are being scattered off of an initial tachyon repeatedly.

In figure 1, the scattering of three systems in a mixed setup with two infinite strips with width $L$ and distance $D$ among them has been shown. This figure is the corresponding figure. 1 of [18],

but in a mixed structure and in a confining background with an end wall positioned at $u_{KK}$. The classical chaos can be studied by the model of pinball scattering which is shown in its left part, where the angle of the outgoing particle is very sensitive to the impact parameters relating to several parameters of the system. This parameter though is not very sensitive to the "quantum" mutual information among the two systems $A$ and $B$.

In the middle plot, the scattering off of a black hole systems is shown where any perturbation in any parameter of this system could cause a large change in the state of the outgoing Hawking quanta with the exponential growth of out-of-time-order correlator (OTOC) and therefore shows the very chaotic nature of the black hole. The level of chaos in this case is higher than the previous case in the sense that it has both "classical" and "quantum" chaos. Note that in this case, depending on the size of the black hole, position of the wall $u_{KK}$ and the sizes $L$ and $D$, various phases could be present.

Then, in the right part, a complicated and highly excited open string structure is positioned in our confining mixed system and the scattering amplitude again shows high level of chaos and high dependence on various parameters of the system, as there is an exponential large number of internal states and dependence on the parameters of the system. Also, based on the size of the parameters of this setup relative to each other and the strength of the correlations among the two systems, in addition to the scattering parameters, one could imagine that at any specific circumstance, a particular saddle would be dominant and therefore a very rich phase structure could be envisioned.

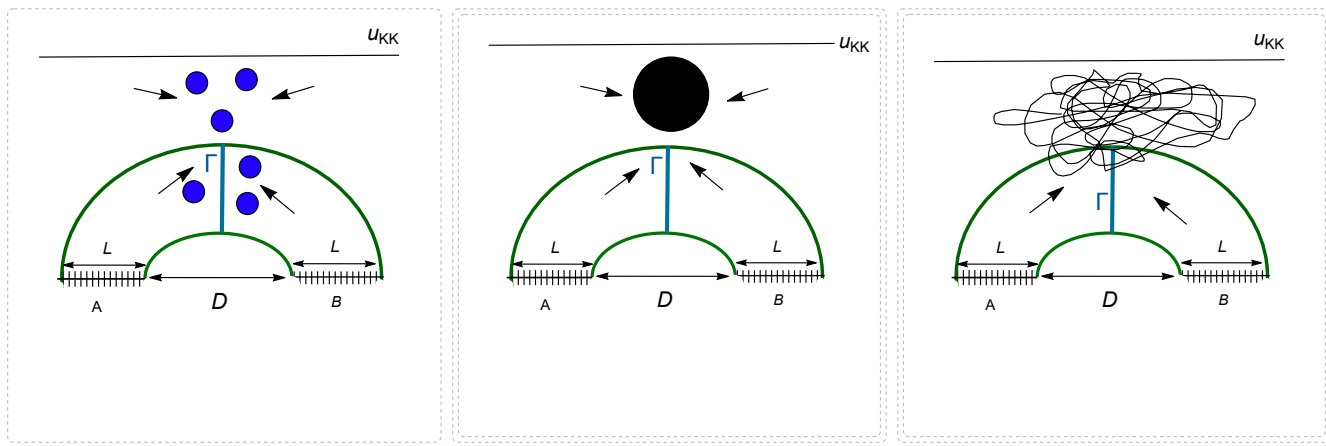

*Figure 1: In the left part, the scattering of classical pinballs, in the middle section the quantum and highly chaotic scattering process from a black hole system, and in the right part the setup for scattering off of highly excited strings are shown.*

As mentioned, this figure actually is related to figure 1 of [18]. However, the difference between our setup and their setup is that we consider the background to be the entanglement wedge which is being constructed from the "entanglement structure" of "two mixed correlated subsystems". One can imagine the modular flows and the internal modular time instead of physical time for this system which describe the pattern of entanglement. Then, in three different setups of [18],

namely classical scattering, a black hole, and a stack of highly excited strings, we tried to depict the interconnections between mutual information and scattering amplitude. One could imagine modular flows from these two sources of mixed subsystems which hit the wall at the end of the geometry and making the peaks which decay in bigger Mandelstam variable "s".

Moreover, one should notice that in quantum field theories, the measure of mutual information is much better than the entanglement entropy as it is UV regulated, and therefore it could be a much better measure to extract the universalities in the structure of entanglement and depict the relations to the intrinsic behaviors of the strings. In addition, as pointed out in [19], compared to entanglement entropy, mutual information is less constrained by the symmetries and therefore it could present better the intricate information about the fundamental theories, such as the full spectrum of the CFTs that is directly related to the behaviors of strings.

One should also remember that already, the entanglement entropy has been employed to find various connections between energy conditions and information theory. For example in [20], using the second order shape deformation of the geometric entanglement entropy, a conjectured lower bound on the expectation value of components of energy momentum tensor has been found which is dubbed the Quantum Null Energy Condition (QNEC). In addition, there, it has been emphasized that in general the properties of modular Hamiltonian under shape deformations for any state $\psi$ is an important quantity to be considered. In that direction, modular Hamiltonian and modular flow have been used extensively [20, 21].

Specifically, we noticed that the shapes of the singularities in the behavior of scattering amplitudes, observed in [1], that form logarithmic branch cuts (rather than poles), which are more pronounced in small $s$ and fade away in the larger $s$, are indeed correct and physical and can be explained by the connections with entanglement entropy and mutual information, as we have observed numerically the same structures in our previous works on mutual information in confining geometries [2, 22].

In [1], using holographic confining backgrounds, the partonic behavior of scattering processes at large and medium ranges of Mandelstam variable $s$ have been explored, where the branch cut singularities at smaller values of $s$ have been detected, whereas the peaks decline at large $s$. This work was based on the previous works of Polchinski and Strassler [23], whose main result was writing the $4d$ amplitude in terms of the $10d$ scattering amplitude as

$$\mathcal{A}_4(s,t,u) \propto \int dr \sqrt{-g} \times \mathcal{A}_{10}[\tilde{s}(r), \tilde{t}(r), \tilde{u}(r)], \tag{1.1}$$

where $\mathcal{A}_{10}$ is the $10d$ string amplitude, $r$ is the holographic radial coordinate, and $\Psi(r)$ is the wave function of the scattered state. In the holographic QCD models with a non-trivial dilaton field such as Maldacena-Nunez [24] or even Witten-QCD [25, 26], where the dilaton field depends on

the radial coordinate, the above relation should instead be written as

$$\mathcal{A}_4(s,t,u) = \int_{u_{KK}}^{\infty} du \sqrt{-g} e^{-2\phi} \Big( \prod_{i=1}^{4} \psi_i(u) \Big) \mathcal{A}_{10}[\tilde{s}(r), \tilde{t}(r), \tilde{u}(r)]. \tag{1.2}$$

Then, there, the amplitudes has been expanded around its poles and it has been shown that the singularities, instead of being the actual poles, would be in the form of branch points or branch cuts with finite imaginary parts (which is the sign of bound state creation) and therefore, the authors of [1], suggested that the prescription of Polchinski and Strassler [23] would fail in this limit. In these scenarios, one might think that other tools such as new holographic mixed quantum information and correlation measures, such as mutual information, negativity or entanglement of purification could be new tools to probe these cases and give new information.

In order to reinforce this, we propose that there is a strong connection between the behavior of entanglement entropy and quantum correlation measures among two mixed subsystems at various energies and scales of the setup, and the behaviors of string scattering amplitudes in the holographic QCD backgrounds, where the quantized behavior and the creation of bound states could be observed in both cases. This connection has been found using these three works [1, 2, 4], and based on the numerical studies performed in the confining geometries.

A main point is that the spectrum of QCD, unlike theories such as $\mathcal{N} = 4$ SYM, is discrete, which is the result of the IR end-wall and the boundary condition that it imposes. This discrete spectrum then can be caught by the quantum information correlation measures. In addition, as found in [1], scattering amplitude as a measure, could catch the transition from soft to hard scattering, which is the "bending" trajectories in the confining models. Moreover, entanglement entropy and mutual information can detect the effects of asymptotic freedom where the gauge couplings and the binding energies become smaller. At lower energies they could also detect the "quantized" behaviors of such binding energies.

It is worth to notice that most of the closed strings reside near the wall which causes the wave function to be peaked at the wall as well, instead of vanishing there, which might seem counterintuitive. This accumulation also could show an effect on the entanglement patterns and mixed correlations saddles, specially, this would be more pronounced at lower energies. Therefore, due to the connections between the partonic nature of hadron scattering and holography [1, 10, 27], one would expect that the entanglement patterns and these scattering structures would have strong connections among them as well, which we can now show here. Additionally, due to the relations between chaos of excited strings and back holes, one would expect the best physical quantity to check first would be the scattering amplitude as the perturbative string theory has been formulated using scattering amplitude.

This paper is organized as follows. In section 2, we discuss the fractal structure coming from the phase transitions and chaos in confining geometries due to the presence of the wall which has been

captured by the mutual information and critical distance $D_c$ in our previous works [2,22,28,29]. In section 3, similar to the results of [1], we explore scattering amplitude in $10d$ and $4d$, the structures of its zeros and poles, and the indication to chaos from their statistics. We also discuss how this zeros' structures behave the same for both mutual information and string scattering amplitude. In section 4, the mutual information and modular flow as an intermediary concept has been used to further establish the relations between mutual information and string scattering amplitude. In section 5, we show how mutual information itself can detect chaotic behaviors. There, we also discuss quantum scattering on a leaky torus and using this specific geometry, we discuss further the chaotic behaviors of strings and also the chaotic spread of mutual information in confining geometries. In section 6, we analyze two examples of Compton scattering with a witness and also the decay of a highly excited string into two tachyons and for each case we discuss the behavior of entanglement entropy or mutual information during the process and so could look into the proposed connection in more details. Then, in section 7, as another example of chaotic behavior in confining geometries, we considered the kink-kink and kink-antikink scattering, which similar to the behavior of $D_c$ and mutual information produce a periodic behavior in the scattering and then we discussed the connections with the previous sections. Finally, in section 9, we summarized the paper with a short conclusion.

## 2 Critical distance in confining backgrounds

First, turning to the EE side, in [30,31], it has been shown that for a general confining background with the geometry of

$$ds^2 = \alpha(u)[\beta(u)du^2 + dx^\mu dx_\mu] + g_{ij}d\theta^i d\theta^j,$$
$$u_{KK} < u < \infty, \quad x^\mu(\mu = 0, 1, ..., d), \quad \theta^i(i = d+2, ...9), \tag{2.1}$$

the length of a strip and the corresponding Ryu–Takayanagi surface, in terms of the minimum of the holographic radial coordinate, $u_0$, can be written as

$$L(u_0) = 2 \int_{u_0}^{\infty} du \sqrt{\frac{\beta(u)}{\frac{\mathtt{H}(u)}{\mathtt{H}(u_0)} - 1}},$$
$$S(u_0) = \frac{V_0}{2G_N} \int_{u_0}^{\infty} du \sqrt{\frac{\beta(u)\mathtt{H}(u)}{1 - \frac{\mathtt{H}(u_0)}{\mathtt{H}(u)}}}, \tag{2.2}$$

where here $\mathtt{H}(u) = e^{-4\phi}V_{\text{int}}^2\alpha(u)^d$ and $V_{\text{int}} = \int d\vec{\theta}\sqrt{\det[g_{ij}]}$. The corresponding plot is shown in figure 2.

Then, for two strips of width $L$ and the distance $D$ between them, the mutual information

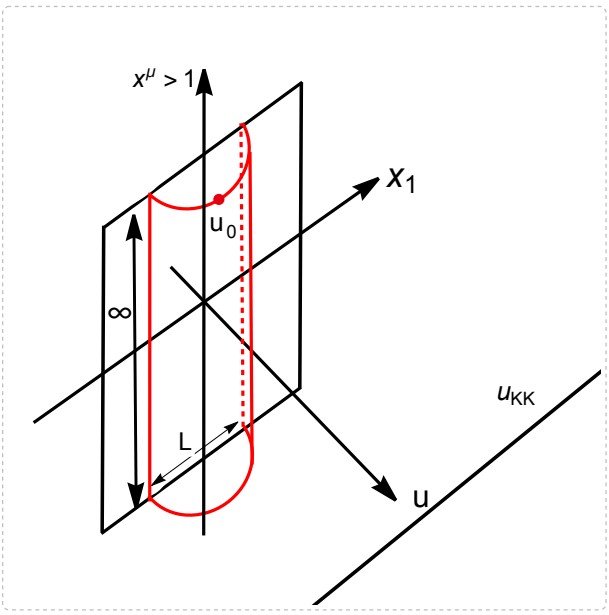

*Figure 2: The setup of one strip and its corresponding Ryu-Takayanagi surface in a confining background.*

would be

$$I(D, L) = 2S(L) - S(D) - S(2L + D), \tag{2.3}$$

and the critical distance $D_c$ is where $I(D_c, L) = 0$. This quantity can be noted in our figure 1 here where the two mixed systems are two strips in a confining background with the end wall at $u_{KK}$.

This quantity, $D_c$, has been implemented in our previous works [2, 22, 29, 32, 33] to probe the quantum correlation properties of various geometries.

In [2], we noticed that, in the background of confining geometries, the critical distance between two strips, which is related to mutual information and in fact is a measure of mixed correlations, shows singularities at smaller values of $u_{KK}$ and statistically smooths out in the bigger values of $u_{KK}$. This behavior is especially pronounced in the Witten-QCD geometry, figure 19 of [2], which is shown again in figure 3. The same singularity jumps can also be seen in the background of Klebanov-Tseytlin geometry [34, 35], shown in figure 4.

Similar to the results of [1], it could be seen that the peaks of $D_c$ which correspond to the peaks of mutual information and mixed correlation measures are stronger and more noticeable in the medium (or rather smaller) regions of $u_{KK}$ which itself corresponds to small values of Mandelstam variable $s$. For bigger $s$ (and bigger $t$ where the angle is fixed), corresponding to bigger values of $u_{KK}$, the peaks would fade away. This is due to the effects of asymptotic freedom where the gauge couplings and the binding energies of the bound states are small. However, from figures 3 and 4, it could be seen again that there is no peak at larger $u_{KK}$, which is due to the fact that the binding energy, length and mass of the strings become smaller in that regimes too.

The connection between critical distance $D_c$ and $u_{KK}$ in the background of Sakai-Sugimoto [36]

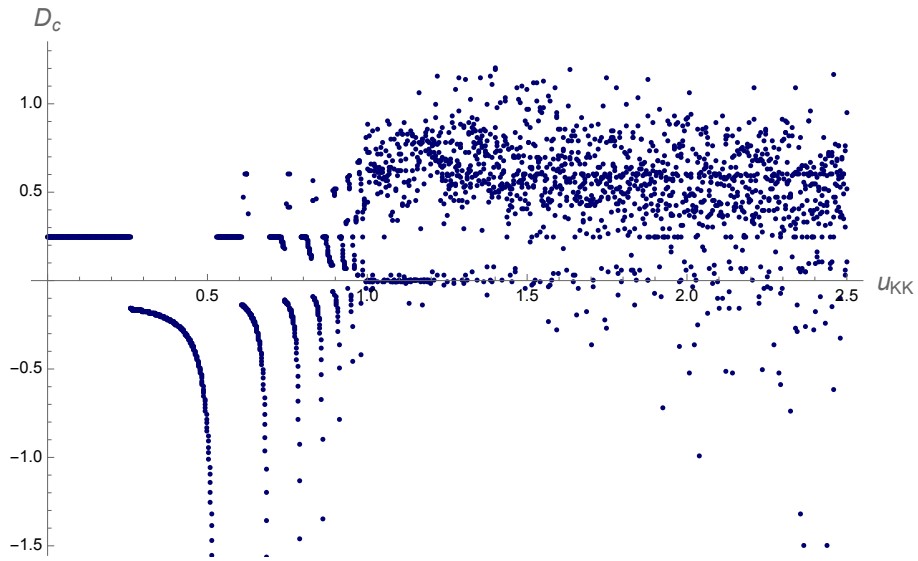

*Figure 3: The behavior of the critical distance, $D_c$, which comes from the mutual information between two mixed strips, versus the position of the IR wall, $u_{KK}$, in the Witten-QCD background. This figure can simply be compared with the figure 6 of [1], for the "real parts of the amplitude", as one can see the behavior of branch cuts and then the decay at large $s$ are indeed very similar to the pattern of MI and $D_c$ in this model. This is the main observation of this section.*

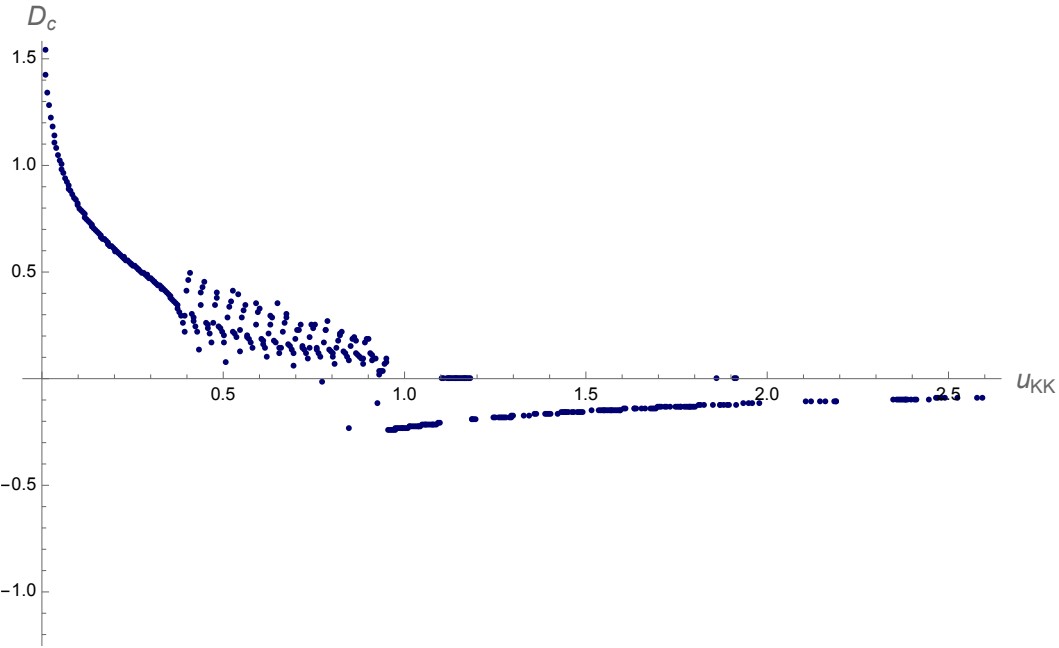

*Figure 4: The relationship between the critical distance $D_c$ among two strips, versus $u_{KK}$, in the background of Klebanov-Tseytlin geometry.*

is shown in figure 5. As this figure has been created from the real part of entanglement entropy, i.e, $S(u_0)$, versus the real part of the size of the strips $L(u_0)$, one would expect that, similarly in the "real" part of the amplitude, a power low decay behavior should be observed at bigger values of $s$ or $u$, as indeed one can see such behaviors from figure 5. The relation for amplitude follows

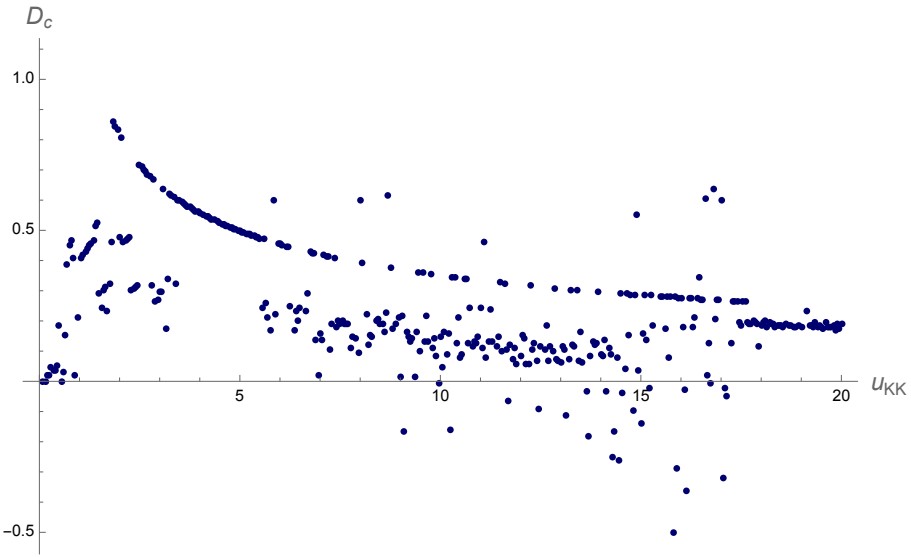

*Figure 5: The relationship between the critical distance $D_c$ among the two strips versus $u_{KK}$, in the background of Sakai-Sugimoto geometry. Again, one can see that the peaks become less pronounced as $u_{KK}$ increases and they fade away at large values of $s$. The decay at large $u_{KK}$ is also a power law.*

$\mathrm{Re}[\mathcal{A}_4] \sim s^{-1}$ and $\mathrm{Im}[\mathcal{A}_4] \sim s^{2-\Delta/2}$, where $\Delta$ is the sum of the scaling dimensions of the operators that contribute as $\prod_{i=1} \psi_i(r) \sim r^{-\Delta}$, ($\Delta$ can be replaced by the twist in QCD models).

The transition from regular to chaotic can be seen clearly in the behavior of $D_c$ in the background of AdS-Soliton shown in figure 6. This plot should be compared with figure 7 of [37] where the transition from regular to random can be seen at larger values of $u_{KK}$s. Bigger values of $u_{KK}$ correspond to smaller values of the discrete levels or "eigenvalues". The peaks become less pronounced as $u_{KK}$ increases, and fade away at large values of $s$. This is similar to the case of [37] which as energy or $s$ increases, the singularities would fade away. This decay at large $u_{KK}$ is also a decreasing power law similar to the relation for $s$. In [37], the interpretation for the large energy case has been pointed out to be the asymptotic freedom as the asymptotic freedom is being approached, the corresponding gauge coupling becomes weaker and therefore the binding energy of the bound states becomes smaller. This is also the case here where as $u_{KK}$ increases, one reaches to asymptotic freedom, the binding energy decreases which based on first law of entanglement entropy would lead to smaller peaks in entanglement entropy, mutual information and also $D_c$.

For interpreting this results, one can visualize the spread of some kinds of waves between the two entangled strips in spacetimes where the "modular waves" interfere with each other and then collide with the wall at the end of the confining geometry and ripple back, creating chaos and then fade away at bigger values of $s$.

One could imagine modular waves being sourced from the two entangled mixed subregions which being hit the end-wall and then getting scattered back, creating modular chaos which form the specific peak structures in the plots of mutual information that we have detected here.

Note that it has been demonstrated, for instance in [38], that the modular chaos as in the case

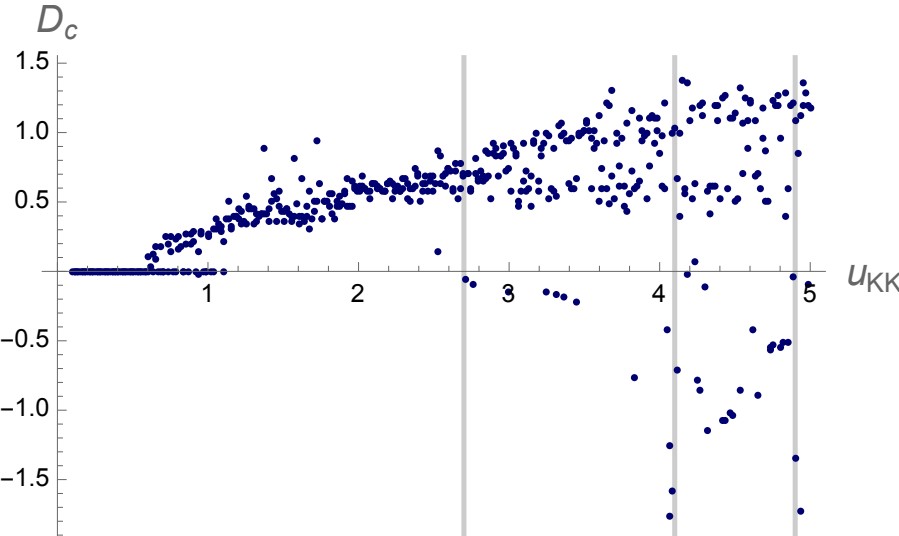

*Figure 6: The relationship between the critical distance $D_c$ among the two strips versus $u_{KK}$, in the background of AdS-soliton geometry. From this figure one can notice that as the position of the end wall, $u_{KK}$, increases, the peaks become less pronounced.*

of entanglement (and mutual information) can reconstruct the holographic bulk geometry. In [1], the connections between chaos and peak structures of string amplitude have been discussed. So, the peak structures of string amplitude and entanglement structures should be connected with each other through chaos and specially the "modular chaos", as we have demonstrated here.

# 3   String scattering amplitude in confining backgrounds

Now turning to the string amplitude side, the $10d$ amplitude in [1] has been found to be

$$\mathcal{A}_{10} = 4^4 \sum_{n=0}^{\infty} (-1)^n \frac{(n+1)^2}{(n!)^2} \frac{(1 + c_+^4 + c_-^4)}{\alpha' s/4 - (n+1)} \times$$

$$\frac{\Gamma\left((n+1)c_+\right)\Gamma\left((n+1)c_-\right)}{\Gamma\left(1 - (n+1)c_+\right)\Gamma\left(1 - (n+1)c_-\right)} \equiv \sum_{n=0}^{\infty} \frac{\mathcal{R}(\theta)}{\alpha' s/4 - (n+1)}, \tag{3.1}$$

where $c_\pm = \frac{1}{2}(1 \pm \cos\theta)$. Higher $\alpha' s$ corresponds to higher energies and bigger $u_{KK}$.

In terms of $s$ and the angle $\theta$, this amplitude can be written as

$$\mathcal{A}_{10} = \frac{1}{32} \alpha'^4 s^4 (\cos(2\theta) + 7)^2 \frac{\Gamma\left(-\frac{1}{4}(\alpha' s)\right)}{\Gamma\left(\frac{\alpha' s}{4} + 1\right)} \times$$

$$\frac{\Gamma\left(-\frac{1}{8}\alpha' s(\cos(\theta) - 1)\right)}{\Gamma\left(\frac{1}{8}(-\alpha' s + \alpha' s \cos(\theta) + 8)\right)} \times \frac{\Gamma\left(\frac{1}{8}\alpha' s(\cos(\theta) + 1)\right)}{\Gamma\left(1 - \frac{1}{8}\alpha' s(\cos(\theta) + 1)\right)}. \tag{3.2}$$

Its behavior versus the Mandelstam variable $s$ in confining holographic backgrounds such as

Witten-QCD has been shown in [1]. The specific confining backgrounds considered there include the hard wall model, soft wall model and Witten's QCD model, showing the same singularity patterns as the EE and MI as shown in figure 7 here. Then, using the relations 1.1 and 1.2, $\mathcal{A}_{10}$ can be integrated and the scattering amplitude in $4d$ can be derived. The results for different values of $\theta$ are shown in figure 8, as for our main figure, which should be compared with those of $D_c$ for our various confining models.

The reason for the peculiar behavior in generating singularities in the form of branch cuts at low energies, is the dependence of the string tension on the holographic radial coordinate. In their work, one could see that in $10d$ or $11d$ geometries, the amplitude drops faster compared to lower $d$ geometries, which is similar to the smoothing out of the singularities in the mixed-correlation measures in higher dimensions.

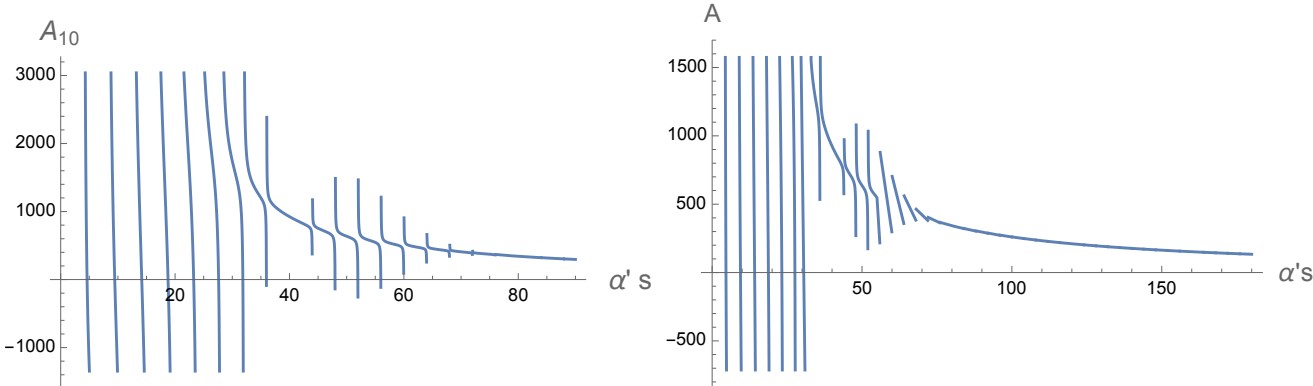

*Figure 7: The relationship between $A_{10}$ versus $\alpha's$ for $\cos\theta = 0.8$. The fading away of the branch cut singularities at large Mandelstam variable s (center of mass energy) is similar to the case of $D_c$ coming from mutual information in confining backgrounds.*

The connections between $s$ and $\theta$ have also been further investigated in figure 9. Note that, there, in the interval $0 < \theta < 3\pi$, nearly 4 branch cuts can be observed, and in the interval of $0 < \theta < 6\pi$, nearly 7 branch cuts can be found. Note that since we are in $10d$ we are allowed to consider such values for the angle $\theta$ as we are considering scattering from highly excited strings in $10d$. Also, since we can consider more spins in the structures of chaotic and excited strings from the reference frame, and specifically since we are in higher dimensions, it would be necessary to consider bigger values than $2\pi$ for $\theta$. In general, with the upper bound of $n\pi$, $n+1$ branch cuts would be present. Also, it can be seen that for bigger $s$, the branch cuts become dispersed, which is also the case for mixed correlation quantum information measures such as mutual information and $D_c$.

One could see that in the simple case of fixed $t$, $\Delta S_E \propto (\log s)^2$, as shown in figure 10. For the mixed correlation measures and in the presence of a hard wall which creates the "four" main saddles [29, 39], the behaviors become more interesting and rich, as shown in figures 3 and 4. So as the parameter $D_c$ which comes from the mutual information is related to the sum of several

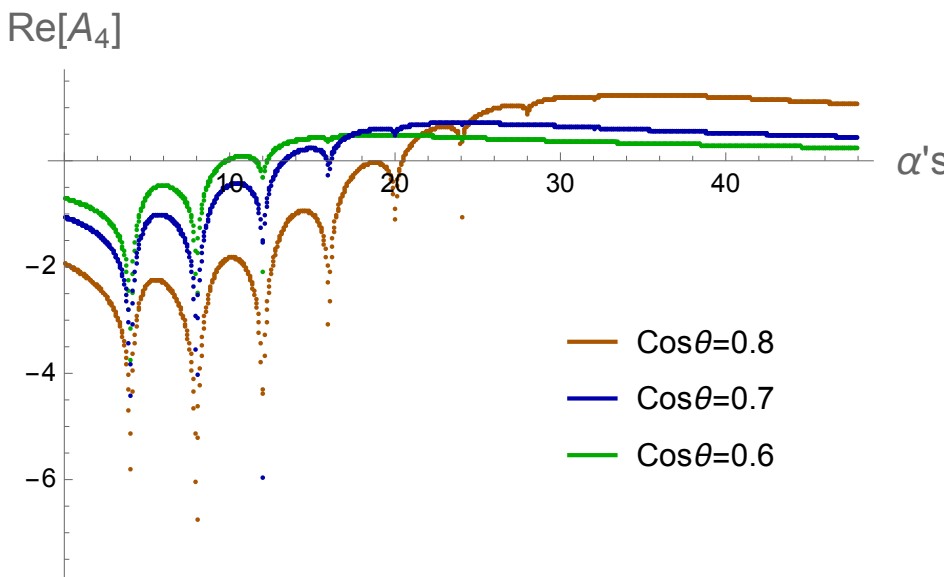

Figure 8: *The relationship between the real part of $[\mathcal{A}_4]$ versus $\alpha's$ using the result of [1] for different values of $\theta$, and in the hard wall confining model. The main behaviors are similar to the case of Witten-QCD. The logarithmic branch cut singularities at lower $s$, and the damping at larger $s$, as $[\mathcal{A}_4] \sim s^{-1}$, are similar to the general behaviors of the mutual information (MI) and critical distance $D_c$, observed in the confining backgrounds.*

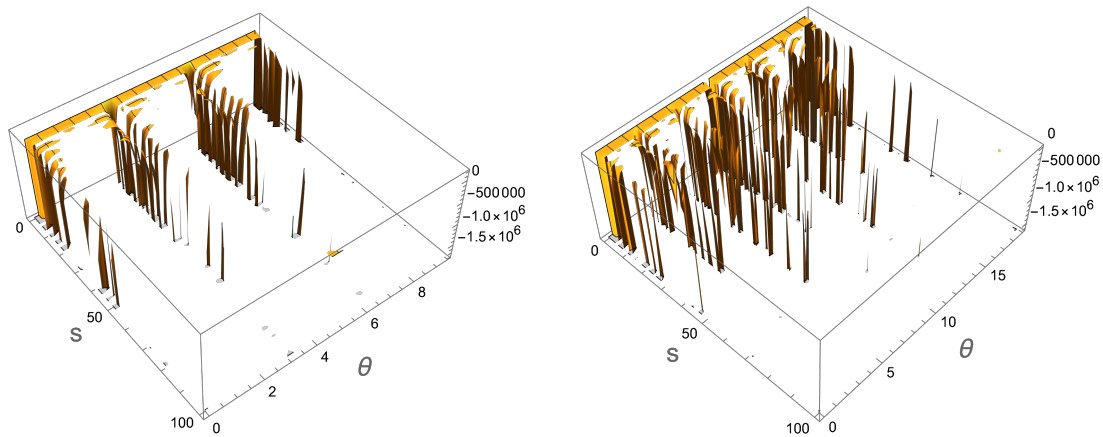

Figure 9: *The 3d plot of the 10d scattering amplitude, $A_{10}$, versus $s$ and $\theta$. We set $\alpha' = 1$ here. In the left the interval for the variable $\theta$ has been chosen to be $0 < \theta < 3\pi$ and in the right it is $0 < \theta < 6\pi$, so in the left four peak can be detected and in the right, seven peaks can be noted.*

entanglement entropies, therefore, figure 10 is comparable to the case of the confining and mixed setup of figures 3 and 4.

It is worthwhile to mention that in [1], the fixed angle limit approximation where $s \to \infty$ while $s/t$ is fixed, has also been studied, where the behavior of the real part of the $A_{10}$ is shown in figure 11. Note that, this limit is not physical as there is in fact imaginary contributions and therefore this case is not related to the quantum mixed correlation measure which we have seen in the full physical $A_{10}$ case without this approximation as well.

The case of the Regge limit, where $s \to \infty$ while $t$ is fixed, gives figures 12, where again the full

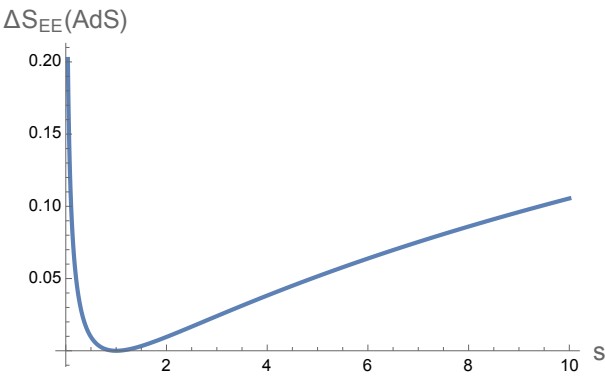

Figure 10: The relationship between $\Delta S_{EE}$ and $s$ in pure AdS.

phase diagram cannot be observed in this setup and consideration of the full scenario is needed. This case then can only detect two main saddles.

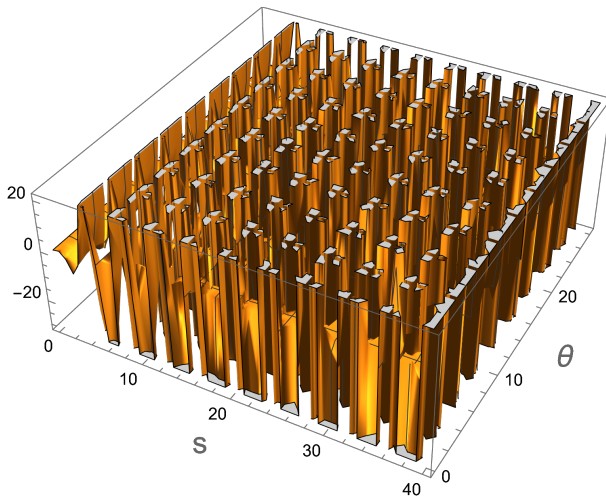

Figure 11: The relationship between $A_{10}$ versus $s$ and $\theta$ for $\alpha = 1$ in the fixed angle approximated limit.

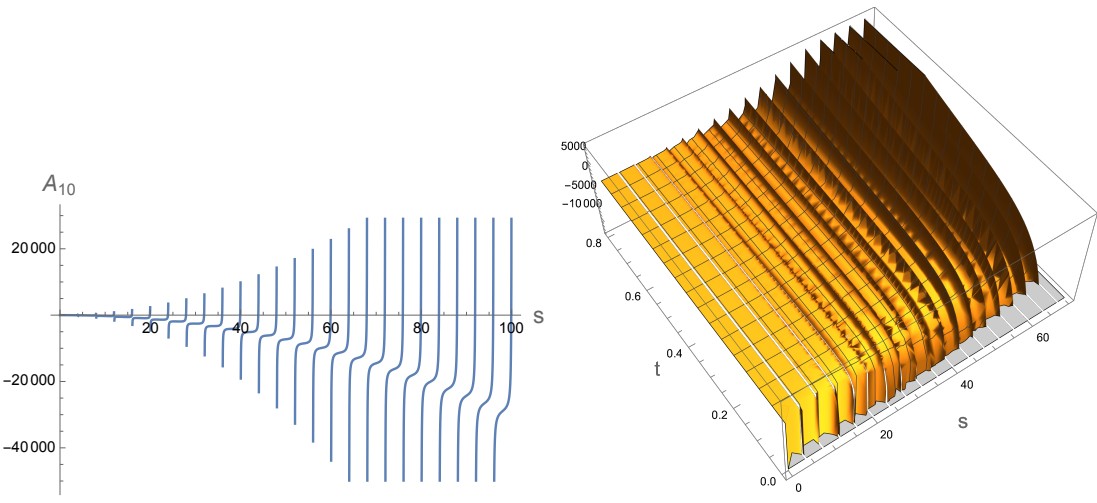

Figure 12: The relationship between $A_{10}$ versus $s$ and $\theta$ for $\alpha = 1$ in the Regge limit.

Note that the scattering process will always be dominated by the interior regions which is close to the IR wall at $u_{KK}$, or $u_\Lambda$, and therefore this parameter should be the main factor in determining the saddle. The mixed quantum correlations would also be more dominant close to that wall leading to the result that the phase transitions can be probed there rather effectively.

Another point is that in [4], by employing the perturbation theory, the disc amplitude of the scattering of two heavy quark-anti-quark pairs which are accelerating in the opposite directions has been studied. Then, using that the entanglement entropy of the quark and anti-quark has been found approximately as $S = \sqrt{\lambda}$, where $\lambda$ is the 't Hooft coupling, the connections between these quantities could firmly be indicated.

Also, in reference [3], it has been shown that as any interaction between particles would change the entanglement entropy (EE), the change of EE should be directly related to the scattering amplitude. This is also the case for all other mixed correlation measures such as mutual information and negativity. So the scattering amplitude should also be able to track the four saddles of mixed correlations [29, 39], as we have noticed. The entanglement entropy can be written in terms of a Wilson loop $\langle W \rangle$ as $S_E = (1 - c\lambda \partial_\lambda) \log\langle W \rangle$, and thus at large 'Hooft couplings $\lambda$, the change of EE at leading order in $\lambda$ is related to the gluon scattering amplitude as

$$\Delta S_E \sim \frac{(1 - \frac{1}{2}c)\sqrt{\lambda}}{8\pi} \left( \log \frac{s}{t} \right)^2. \tag{3.3}$$

Similar connections between the mixed correlation measures such as mutual information, negativity, entanglement and complexity of purification (EoP and CoP) on one hand, and the string scattering amplitudes on the other hand could be foreseen. The specific structure for each measure and in each specific background can be found in future works.

Another interesting observation is the links between the change of EE, or other correlation measures, and the Bremsstralung of radiative corrections as $\lambda \partial_\lambda \log\langle W \rangle$ in the case of accelerating quark-antiquarks pair, which is proportional to the Bremsstralung function. So the spikes that we noticed during the phase transitions in the plot of $D_c$ could be considered as the K-lines in Bremsstralung radiation.

Bear in mind that our observation for the connections is only for closed strings (glueballs). In the case of open strings, which describe mesons or baryons, there would be two kinds of entanglement entropy, which makes finding the connections between mutual information and scattering amplitude more difficult. The EE there includes shares from the entanglement of strings endpoints in the gluons (or flavor branes) and the entanglement between the gluons themselves, making the framework more complicated.

# 4 Mutual information, modular Hamiltonian and binding energy

Entanglement entropy and mutual information have already been used to extract the universalities in the properties of renormalization group flows in various energies [40,41]. Therefore, it is expected that it can also detect the quantized behaviors in the low energies, occurring due to the creation of binding energy. However, direct connections between mutual information and string amplitude has not been found. Here, we can use the modular Hamiltonian, modular flows and shape dependence similar to the setup of [19], to formulate this connection. We only need to model the scattering process and its effects on modular flow as a shape perturbation similar to the setup that has been considered in [19]. Additionally, we can show that modular flows could also detect the partonic behaviors at low energies that has been found in [1].

We can start by using the formula of the mutual information between two spherical regions with radius $R_A$ and $R_B$ separated by a large distance $L$, which follows the relation [19, 42–44]

$$I_{A,B} = \mathcal{N}_\Delta \frac{\sqrt{\pi}\Gamma(2\Delta + 1)}{4\Gamma(2\Delta + \frac{3}{2})} \left(\frac{R_A R_B}{L^2}\right)^{2\Delta} + ... \tag{4.1}$$

In the above relation, $\Delta$ is the conformal dimension of an internal scalar primary operator that in principle "carries" the mixed correlation between the spherical regions $A$ and $B$. The string scattering process would affect this scalar field, weight of the carriers, and therefore this correlation.

The behavior of the mutual information $I_{AB}$ versus $\Delta$, for the case of $L > R_{A,B}$ has been shown in figure 13. From this figure, one can see that by increasing $\Delta$, the mutual information $I_{AB}$ would decrease. Also, as one would expect, for any specific value for $\Delta$, the smaller distance $L$ corresponds to bigger mutual information $I_{AB}$ between the two spheres.

The relation between the scaling dimension and the mass of the field $\phi$ is $\Delta = \frac{d}{2} + \sqrt{\frac{d^2}{4} + m^2 R^2}$. So, for the case of $L > R_{A,B}$, bigger $\Delta$ actually means bigger mass, $m$, for the field, which makes the scattering process between the strings to be done with more difficultly, leading to smaller scattering amplitude. This can also be seen from figure 3 of [45], where the open string profile at different average mass and their time evolution have been shown and one can easily see that by increasing mass, the profile shrink, making the string scattering amplitude to decrease. So the fact that scattering amplitude and mutual information behave similarly with respect to the conformal dimension of the background scalar field, or the mass, is another sign on how they are related with each other, and using one of these quantities, one can measure the other one.

Note that in other models in CFT such as $O(N)$ model, when the scaling dimension increases the system becomes more stable against the fluctuations of the magnetic field or the electric field. The charge $Q$ in these models actually increases the anomalous dimension $\Delta$ [46], and also breaks

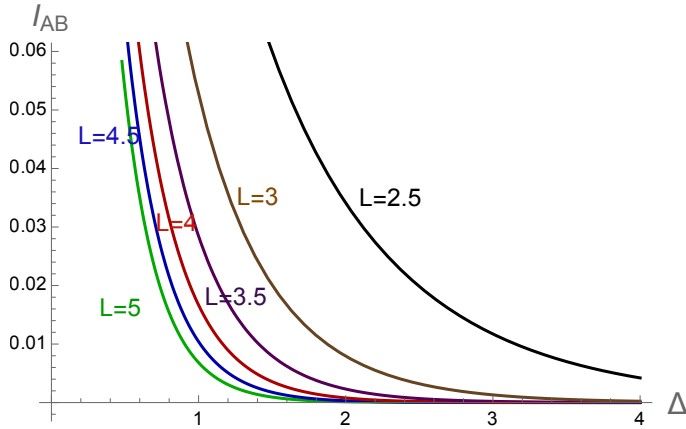

*Figure 13: The relationship between mutual information $I_{AB}$, versus the conformal dimension of an internal scalar primary operator $\Delta$, for various distance $L$ between the spherical regions. As one would expect, for any specific mass or $\Delta$, increasing $L$ would decrease MI, and also at any specific $L$, increasing $\Delta$ would decrease MI. The same behavior is expected for string scattering amplitude versus mass.*

the correlation in a decreasing power law function as in the form of 13.

We can then model the scattering process of an accelerating string or quark moving toward a sphere or another sitting quark by the perturbative model of shape deformation and the modular flow [19]. First, note that for any Hamiltonian $H$, the thermal density matrix can be written as $\rho_\beta = \frac{e^{-\beta H}}{\text{tr}(e^{-\beta H})}$ and from the other side, for any invertible density matrix $\rho$, a thermal Hamiltonian can be constructed as $K_\rho \equiv -\log \rho$, where $K_\rho$ is actually the modular Hamiltonian of the state $\rho$. So, any density matrix can be written as an exponential form of $\rho_V = ce^{-\mathcal{H}}$.

The interesting thing about modular theory of these operators or *Tomita-Takesaki theory* [47–49] is that it can be constructed even for the case where the density matrices cannot be defined for the quantum systems. Note that the modular Hamiltonian actually encodes the complete entanglement data.

For any Hilbert space $\mathcal{H}$ with a von Neumann algebra $\mathcal{A}$ and for a state $|\Omega\rangle \in \mathcal{H}$, the antilinear operator $S_\Omega$ can be defined as [50]

$$S_\Omega(a\,|\Omega\rangle) = a^\dagger\,|\Omega\rangle\,, \qquad a \in \mathcal{A}. \tag{4.2}$$

Using this $S_\Omega$, the modular operator $\Delta_\Omega \equiv S_\Omega^\dagger S_\Omega$ can be defined and then using $\Delta_\Omega$, the modular Hamiltonian can be redefined as $K_\Omega \equiv -\log \Delta_\Omega$. Then, the *modular flow* of operators can be defined via a map as

$$a \mapsto e^{iK_\Omega t}ae^{-iK_\Omega t} = \Delta_\Omega^{-it}a\Delta_\Omega^{it}. \tag{4.3}$$

Note that this is a unitary operator which implements an internal time flow. For the case of more than one interval, this evolution is non-local, which causes the advancements in the causal structure and also "teleportation" between the disjoint intervals [51]. This can signal that for con-

necting scattering amplitude and quantum information, one should consider the mixed correlation measures for multi-regions, and therefore it would be better to check the connections between the mutual information and scattering amplitude rather than the entanglement entropy and scattering amplitude. In addition, note that the modular Hamiltonian can be written as the sum of a local part and a non-local part as $H = H_{\mathrm{loc}} + H_{non-loc}$.

Also, note that $H$ can be written using the decomposition in terms of the eigenvectors as

$$H(x,y) = 2\pi \sum_k \int_{-\infty}^{\infty} ds \; s \; \Psi_s^k(y)\Psi_s^{k*}(x), \tag{4.4}$$

where $\Psi(s) \equiv \Psi(x^\mu(s))$, and $s_1$ and $s_2$ are length parameters along the spherical set $V$. Also $\Psi(x)$s are the Dirac fields. This points to the fact that the sum of the scattered eigenvectors from the scattering process could reconstruct the modular Hamiltonian and the trajectories of modular flow could be connected to the trajectories of the scattered particles.

Also, note that the local part of Hamiltonian can be written as

$$H_{\mathrm{loc}} = \pi i \left( 2 \left( \frac{dz(x)}{dx} \right)^{-1} \partial_x + \frac{d}{dx} \left( \frac{dz(x)}{dx} \right)^{-1} \right) \delta(x-y), \tag{4.5}$$

and the non-local part can be written

$$H_{\mathrm{non\text{-}loc}} = -2\pi i \sum_{l, x_l(z(x)) \neq x} \frac{1}{(x-y)} \left( \frac{dz}{dy} \right)^{-1} \delta(y - x_l(z(x))), \tag{4.6}$$

which mixes a finite number of points $x_l(z)$ from each interval, which are related to the peaks of $D_c$ that we noted in the mutual information. So, one can say that the singularities and the zeros in the plots of scattering amplitude and also $D_c$ are related mostly to the non-local correlations.

One can imagine that due to the scattering process, the boundary of one of the subsystems would change as $\tilde{x}^\mu = x^\mu + \zeta^\mu$. The strings that are being created from the vacuum for an instance in time would move along the geodesics which are determined by the linear response of the metric deformation. Then, the scattering process, similar to [19], can be found to be proportional to the deformation and "modular flow", $\nabla^\mu \xi^\nu$, leading to the metric deformation $\delta g^{\mu\nu} = 2\nabla^\mu \xi^\nu$, which then leads to the change of the reduced density matrix as

$$\delta\rho_A = U^\dagger \circ \rho_{\tilde{A}} \circ U - \rho_A \simeq \frac{1}{2} \int \delta g^{\mu\nu} \rho_A \tilde{T}_{\mu\nu}. \tag{4.7}$$

From the first law of entanglement entropy, one can in general say that the linear response of the entanglement entropy is proportional to a contour integral of stress tensor correlator with modular Hamiltonian as [19] $\delta S_A \propto \oint_{z \sim \partial A} dz \langle T(z) H_A \rangle$. Then, under this action, scattering process

or the shape deformation, the entanglement entropy of one sub-region would change as

$$\delta S_A = \int dx \sqrt{g} \, \nabla^\mu \zeta^\nu(x) \langle T_{\mu\nu}(x) H_A \rangle + \mathcal{O}(\zeta^2). \tag{4.8}$$

This equation should be compared with equation

$$\mathcal{A}_4(s,t,u) = \int_{U_\Lambda}^\infty dU \sqrt{g} \, e^{-2\phi} \Big( \prod_{i=1}^4 \psi_i(U) \Big) \mathcal{A}_{10}(\tilde{s}, \tilde{t}, \tilde{u}). \tag{4.9}$$

From these two equations, one can see how the ten-dimensional scattering amplitude and the wave functions that are being scattered are related to the average of the correlation of the energy momentum and modular Hamiltonian. In other word, we claim that in an elastic scattering process we have the relation $\nabla^\mu \zeta^\nu(x) \langle T_{\mu\nu}(x) H_A \rangle \propto e^{-2\phi} \big( \prod_{i=1}^4 \psi_i(U) \big) \mathcal{A}_{10}(\tilde{s}, \tilde{t}, \tilde{u})$.

The string scattering amplitude also can be written in terms of such integrals, as the integration of the wave-functionals. Additionally, this fact can be demonstrated phenomenologically similarly to the case of [52], where it has been shown that the magnetic field can be tuned in a way that for any kinematic setup, the amplitude vanishes. Similar studies have been done for the entanglement entropy in [28, 53], where it has been shown that for any size of subregion, the magnetic field can be tuned to make the entanglement entropy to get vanished.

Equation 4.6 shows that the change in entanglement entropy only picks up the residue of the simple poles of $\langle T(z) H_A \rangle$ as

$$\langle T(z) H_A \rangle = \ldots + \frac{Res}{z - \partial A} + \ldots \tag{4.10}$$

However, the above arguments are for the case of one subsystem. We would like to extend this to mutual information and discuss its connections with scattering amplitude further. For the case of mutual information, as has been found in [19], the relation then can be written as

$$\delta I_{A,B} = \int_{\partial \mathcal{M}} \epsilon \, d\theta \, d^{d-2}\Omega \sqrt{h(\Omega)} \, n^\mu \zeta^\nu \, \langle T_{\mu\nu}(\theta, \Omega) \Delta H_{A,B} \rangle. \tag{4.11}$$

So the linear response of mutual information, $\delta I_{A,B}$, would be encoded in the "susceptibility" of shape deformation as

$$\frac{\delta I_{A,B}}{\delta \xi(X^i)} = (i) \oint_{|z|=\tilde{z}} dz \Big\langle \Big( T_{zz}(z, \bar{z}, X^i) + T_{z\bar{z}}(z, \bar{z}, X^i) \Big) \Delta \tilde{H}_{A,B} \Big\rangle + h.c. \tag{4.12}$$

which picks up the residue of simple poles of $\langle T_{zz} \Delta \tilde{H}_{A,B} \rangle$, and is independent of the radial coordinate, exactly similar to the case of string scattering amplitude and the residue of its poles as have been discussed in [1].

So for an interval region $A \equiv \{-\frac{l}{2} \leq x \leq \frac{l}{2}\}$, the modular flow which makes the change $\partial^\mu T_{\mu\nu}$

along $H_A$, has a specific trajectory, and a contour integral which is related to the trajectory of the strings being created from the vacuum, i.e, $\partial S_A \propto \oint_{z \sim \partial A} \langle T(z) H_A \rangle$ which as mentioned in [19], takes the residue of the simple pole in $\langle T(z) H_A \rangle$ where the poles that are emerging in the integral of the modular Hamiltonian are related to the poles found for the string scattering amplitude $\mathcal{A}_4$.

Similarly, as has been noted, the scattering amplitude, $\mathcal{A}_4$, can be written as the integral of the metric, the shape deformation due to the scattering process and also in terms of the correlation functions between the stress tensor and modular Hamiltonian, i.e, $\langle T_{\mu\nu} \Delta H \rangle$. In terms of the modular flow, it can be written as $\langle T_{\mu\nu} \mathcal{O}(-is_1) \mathcal{O}(-is_2) \rangle$. Therefore, the logarithmic branch cuts observed in the string scattering process and its trajectories could be simulated by the modular trajectories and the zeros in the correlations between stress tensor and modular Hamiltonian. Also, as mentioned in [19], during the shape deformation, the contributing modes are the ones that move the cut-off tube instead of deforming it. Actually, in low energies, these modes are the one that are related to the binding energy.

The connections between the trajectory of a "probe observer" in a wormhole and its infalling worldline to the black hole interior and the SYK modular Hamiltonian and modular flow has also been discussed in [54]. As discussed there, the scattering amplitude in terms of the modular flow can be written as

$$\mathcal{A}\left(\{\phi_l^i, \phi_r^j\} \to \{\chi^k\}\right) = \text{Tr}\left(\rho^{1-is} \chi^1 ... \chi^n \rho^{is} \phi_l^1(t_{l1}) ... \phi_l^i(t_{li}) \phi_r^1(t_{l1}) ... \phi_r^j(t_{rj})\right), \quad (4.13)$$

where $\{\chi^k\}$ is the set of bulk operators which act on a thermofield double state, $\rho$ is the probe and $\phi_l^1(t_{l1}) ... \phi_l^i(t_{li}) \phi_r^1(t_{l1}) ... \phi_r^j(t_{rj})$ is the boundary state which generates the particles. So through this link between scattering amplitude and modular flow, the relations between mutual information and scattering amplitude is also expected and therefore the information that scattering amplitude could catch such as detecting the binding energies at low $s$ could be caught by modular flow, entanglement entropy and mutual information as well. However, as explained before, the correlation with mutual information would be more direct and physical compared with the entanglement entropy.

As the connections between mutual information, or critical distance $D_c$ has been established, one can further give evidence why the authors in [1], observed logarithmic branch cuts. The mutual information between two spheres with radius $R_A$ and $R_B$ can be written as [19]

$$I_n(A, B) = \frac{1}{1-n} \log\left(\frac{\text{tr}\rho_A^n \text{tr}\rho_B^n}{\text{tr}\rho_{AB}^n}\right) = \frac{1}{n-1} \log\left(\frac{Z(\mathcal{C}_n^{A \cup B}) Z^n(\mathcal{M})}{Z(\mathcal{C}_n^A) Z(\mathcal{C}_n^B)}\right)$$
$$= \left(\frac{2R_A 2R_B}{L^2}\right)^{2\Delta} \frac{\sqrt{\pi}}{4^{2\Delta+1}} \frac{\Gamma(2\Delta+1)}{\Gamma(2\Delta+3/2)}. \quad (4.14)$$

The first line here gives another evidence why in [1], in the behavior of string scattering amplitudes, the logarithmic branch cut behaviors has been observed. Note that in the above relation, $\mathcal{M}$ is the Euclidean spacetime manifold where the QFTs have been defined on, and $Z$s are the

partition functions.

The part of the gamma factors in the above relation can be written as $2\Delta\rho^{2\Delta}\frac{\sqrt{\pi}}{4^{2\Delta+1}}\frac{\Gamma(2\Delta+1)}{\Gamma(2\Delta+3/2)} = \left(\frac{\partial I_{A,B}}{\partial \ln\rho}\right)$, and therefore as written in [19], the linear response of the mutual information between the two spherical regions due to the scattering process or any other perturbative effects like shape deformation, could be written as

$$\delta I_{A,B} = N_d \left(\frac{\partial I_{A,B}}{\partial\ln\rho}\right) \int d\Omega_{d-2} \left(\frac{L^2 - R_A^2}{L^2 + R_A^2 - 2LR_A\cos\alpha}\right)^{d-1} \zeta(\Omega). \tag{4.15}$$

This leads to a form of susceptibility which is completely similar to the integral for calculating the amplitude of the scattering process. This is actually due to the fact that the entanglement has a non-local structure. Note that the above relation also points to the fact that such response only depends on the "zero-modes". Also, the above relations are for spin zero which can be extended to higher spins too.

Another point which is related to the interconnections between modular Hamiltonian and string scattering, is the additive behavior in $s_k$ as the response of mutual information can be written as [19]

$$\delta I_{AB} = \int ds_k \text{``}\delta I_{AB}(s_k)\text{''}. \tag{4.16}$$

This additive behavior is actually the result of the deep connections between mutual information, modular flow and string scattering process in the vacuum.

# 5 Chaos, mutual information and string scattering amplitude

The chaotic behavior of highly excited strings (HES) state with two or three tachyons at the end of these open strings has been investigated in [37, 55–58]. There, it has been shown that the peaks in the angular dependence of the amplitude would behave similar to the $\beta$-ensemble of random matrix theory and therefore demonstrated that their behavior would be chaotic. The pattern of the successive peaks in the mutual information in some specific phases which can be seen from figures 3 , 4 and 5, can also point to the chaotic behavior in that phase where the damping of it behaves based on $\beta$-ensemble.

In [59], also it has been discussed that the "tripartite information" of Choi state for the bipartite unitary channels can detect chaos where it becomes negative. In the case of the most negative value of $I_3 = -2\log d$ where $d$ is the prime dimension, the system is maximally scrambled. Now we propose that for two strip subsystems in the background of *"confining models"*, the mutual information also can detect the chaotic behavior which would be present in some specific phases and for specific ranges of the parameters of the model.

Note that the links between tripartite mutual information and chaotic behaviors have also been discussed in [60–62] and the mutual information at the edge of chaos (where the largest Lyapunov exponent of the dynamical system transitions from negative to positive) in reservoir computers has been discussed in [63]. There, also the mutual information between the training signal $g(t)$, and the signal which is being produced from the linear combination of reservoir signals $h(t)$, has been investigated and the effect of this correlation i.e, $I[g(t), h(t)]$ on the performance of reservoir computer have been tested. It has been demonstrated there that the larger mutual information between the fit signal $h(t)$ and the training signal $g(t)$ would lead to a better performance for the reservoir computer. So at the edge of chaos, the mutual information is maximum and the performance is the most optimum. Also, there, the modular flow and the string scattering amplitudes are maximum. In the setup of holography, this connection has been examined in [64].

As noted in [37], where the authors observed chaotic behaviors in the scattering amplitude of highly excited string due to the string/black hole correspondence, the investigation of such chaotic behaviors can unfold the chaotic behavior of black holes as well.

In [37], it has been shown that the interplay between the highly excited strings (HES) dressing factor and the Veneziano amplitude that it multiplies would create the transition from the chaotic to regular spacing as shown in their figure 7. This could be written as

$$\mathcal{A}_{HES}(s,t) = \mathcal{A}_{\text{Ven}}(s,t)\mathcal{D}_{HES}(s,\theta), \tag{5.1}$$

where at the chaotic phase, for the solution of $F_{\mathcal{D}}(\theta) \equiv \frac{d \log \mathcal{D}_{HES}}{d\theta} = 0$, the ratios of consecutive spacing would follow the $\beta$-ensemble distribution. Also, at the edge of chaos, the peaks of the dressing factor $\mathcal{D}_{HES}$ is a complicated product of polynomials where the peaks would be spaced erratically.

Here, in the response of mutual information, by the change in the critical distance, we could also detect the transition from regular to chaotic behavior as shown in our figure 6. Such dressing factor therefore are correlated with the behavior of the mutual information, where at the transition point from regular to chaotic behavior, both behave similarly. One just needs to find some patterns based on random matrix theory (RMT) for the mutual information, perhaps an angular functional of it, in order to establish the link more firm.

In [65, 66], the dressed infrared quantum information and dressed-state formalism have been studied and connections with scattering and S-matrix and also black hole information paradox have been mentioned. This dressing effect are actually by the long-wavelength photon modes.

The dressing factor in [65] has been determined by first defining the photon soft factor

$$F_\ell(\mathbf{k}, \mathbf{p}) = \frac{p \cdot e_\ell(\mathbf{k})}{p \cdot k} \phi(\mathbf{k}, \mathbf{p}), \tag{5.2}$$

where $\mathbf{k}$ is the photon momentum, $\ell = 1, 2$ would label the photon polarization states and $\phi(\mathbf{k}, \mathbf{p})$

is any function that asymptotes to one when the momentum vanishes.

Then, for a single-electron, the dressing operator is [65]

$$W_{\mathbf{p}} = N_{\mathbf{p}} \exp\left\{ e \sum_{\ell=1}^{2} \int \frac{d^3\mathbf{k}}{\sqrt{2k}} F_\ell(\mathbf{p}, \mathbf{k}) a_\ell^\dagger(\mathbf{k}) \right\} \times \exp\left\{ - e \sum_{\ell=1}^{2} \int \frac{d^3\mathbf{k}}{\sqrt{2k}} F_\ell^*(\mathbf{p}, \mathbf{k}) a_\ell(\mathbf{k}) \right\}, \qquad (5.3)$$

where the normalization factor is

$$N_{\mathbf{p}} = \exp\left\{ -\frac{e^2}{2} \sum_{\ell=1}^{2} \int \frac{d^3\mathbf{k}}{2k} |F_\ell(\mathbf{P}, \mathbf{k})|^2 \right\}. \qquad (5.4)$$

Here, $\lambda$ is the photon mass which acts as the IR regulator and $E > \lambda$ is the energy resolution of a single-photon detector which is the infrared cutoff. So one can see that for $\phi$, $e_\ell$ and smaller momentum $k$, the dressing factor is bigger. Also, the dampening factor could be written as [65]

$$D_{\mathbf{q}\mathbf{q}'} = \exp\left\{ -\frac{e^2}{2} \sum_{\ell=1}^{2} \int \frac{d^3\mathbf{k}}{2k} |F_\ell(\mathbf{q}) - F_\ell^*(\mathbf{q}')|^2 \right\} \qquad (5.5)$$

Note that the low-energy behaviors of photons are important in the process. As mentioned in [37], an example of a $\beta$-ensemble system is the log-gas, where its eigenvalues have the probability density function of

$$P_N(\lambda_1, \lambda_2, ..., \lambda_N) = \mathcal{C}(\beta) \times \exp\left( -\frac{\beta}{2} \sum_{i=1}^{N} \lambda_i^2 \right) \prod_{1 \leq i < j \leq N} |\lambda_i - \lambda_j|^\beta, \qquad (5.6)$$

where $\beta = 1$ for Gaussian orthogonal ensemble, $\beta = 2$ for Gaussian unitary ensemble or $\beta = 4$ for Gaussian symplectic ensemble, and $\mathcal{C}(\beta)$ is a normalization constant.

On the other hand, in [67], the distribution of mutual information of multiple-input multiple-output (MIMO) channel can be modeled by our two equal and symmetric strips. They could show that the distribution follows a Gaussian approximation based on Coulomb fluid similar to the case of $\beta$-ensemble of random matrices for scattering amplitude where the Coulomb gas would be an example of it as well [37].

Also, in [67], it has been shown that the deviations from Gaussian in the distribution of mutual information are affected by the parameters such as the number of interferers and the signal-to-interference-ratio. These parameters would similarly affect the distribution of energy levels of the string scattering amplitude, the spacing and also the ratios of consecutive spacings, as discussed in [37]. In [37], it has also been shown that in the case of $3 \times 3$ matrix which has the eigenvalues $\lambda_1 < \lambda_2 < \lambda_3$, in the $\beta$-ensemble, the probability density function of the ratio $r = (\lambda_3 - \lambda_2)/(\lambda_2 - \lambda_1)$

is

$$f_\beta(r) = \frac{3^{\frac{3+3\beta}{2}}\Gamma(1+\frac{\beta}{2})^2}{2\pi\Gamma(1+\beta)} \frac{(r+r^2)^\beta}{(1+r+r^2)^{1+\frac{3}{2}\beta}},$$

(5.7)

which is similar to the probability distribution function of normalized mutual information of MIMO system and energy levels of a Coulomb fluid. The plot of distribution is shown in figure 14. It can be seen that by increasing $\beta$, the maximum in the distribution function becomes bigger and get further away from each other.

The conditional mutual information between the input and output channels is also [67]

$$I(\mathbf{x};\mathbf{y}) = \log\det\left(I_{n_r} + \frac{P}{P_I/k}\mathbf{H}\mathbf{H}^\dagger(\mathbf{H}_I\mathbf{H}_I^\dagger)^{-1}\right).$$

(5.8)

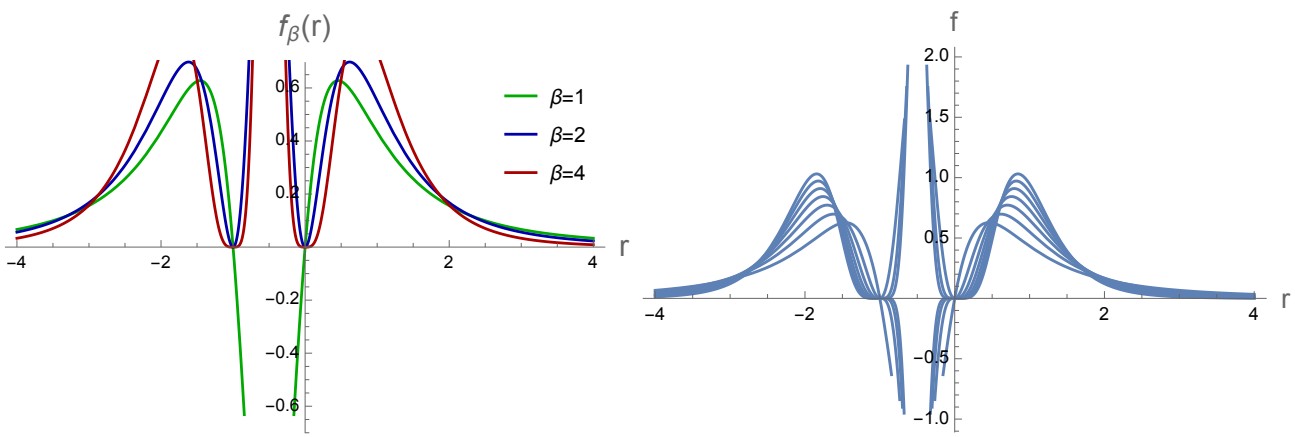

Figure 14: The probability density functions for the ratio of eigenvalues in the $\beta$-ensemble which will be used as fitting models.

One should note that all these chaotic behaviors and partonic structures we observed in the scattering amplitude and mutual information originated from the large fluctuations in the spin of the intrinsic highly excited states. The precise effects of the fluctuations in the spin of particles on the quantum information measures specifically mutual information and other mixed correlation measures needs to be studied further.

One method is to check the spacings between successive peaks of mutual information or critical distance $D_C$ as a function of peaks in the position of the hard wall $u_{KKn}$, where in fact similar to [56], the parameter $u_{KK}$ is a continuous kinematic variable as well. As we have observed from our different plots, the spacing there are erratic, similar to the patterns that one would expect from the spectra of chaotic systems such as the place of the zeros of Riemann zeta function which has the distribution close to the random matrix theory in the Gaussian unitary ensemble.

As it has been introduced in [56], the new measure of chaotic scattering amplitude would take the form of a log-normal distribution for the consecutive peaks. This log-normal function could

also be seen in the pattern of the peaks of mutual information versus the critical distance. The interesting point is that the same measure could be applied to the decay of highly excited string state into two tachyons and also the quantum scattering on a "leaky torus". The topology of the leaky torus has been shown in figure 15.

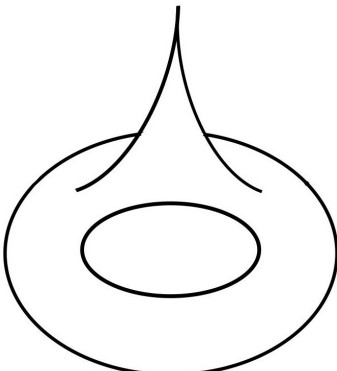

*Figure 15: The topology of a leaky torus embedded in Euclidean 3-dimension space [68]. This geometry is smooth and orientable but not compact, with a constant negative curvature, and an infinite cusp. Also, it contains an open end where the phase shift of a particle which enters through it shows all the aspects of chaos with great difficulty in its effective numerical computation.*

The geometry of the leaky torus $H$ can be written as [68]

$$ds^2 = \frac{dx^2 + dy^2}{y^2},\tag{5.9}$$

with the boundaries of H as

$$(i) \quad x = -1, \quad 0 < y < \infty;$$
$$(ii) \quad (x - \frac{1}{2})^2 + y^2 = \frac{1}{4}, \quad 0 < x < 1;$$
$$(iii) \quad x = 1, 0 < y < \infty;$$
$$(iv) \quad (x + \frac{1}{2})^2 + y^2 = (\frac{1}{2})^2.$$

So, the pattern of the scatterings of the wave-function on the leaky torus and the erratic behavior of its phase shift $\Phi$ as the function of momentum $k$, i.e, $\Phi(k) = \frac{\text{Im}[\zeta(1+2ik)]}{\text{Re}[\zeta(1+2ik)]}$ are directly related to the spreading of information in a confining geometry with a hard wall. The spacings for successive maxima for both the function $\Phi(k)$ and $D_c(u_{KK})$ would be on the zeros on the critical line $\text{Re}[s] = \frac{1}{2}$.

The free wave is being scattered from $y = \infty$ and the phase shift between the incoming and outgoing waves is being measured at $y = y_0 > 0$.

So the problems of quantum state spreading on a leaky torus, scattering of highly excited strings (HES) into two tachyons, and mutual information spreading in the presence of a hard wall

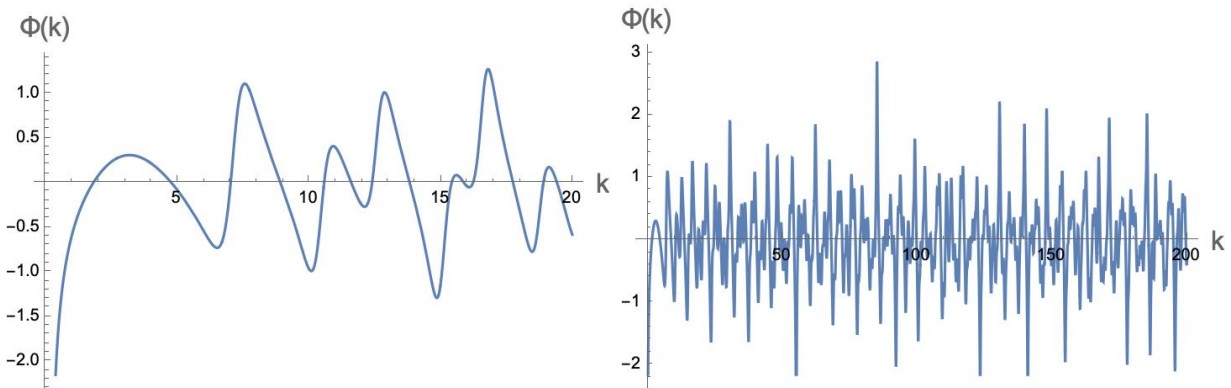

*Figure 16: The chaotic behavior of the phase shift as the function of the momentum of incoming wave on a leaky torus which can be a model of the spread of information in a confining geometry.*

of a confining system are all equivalent, showing chaos in their behaviors, as the peaks of the parameters would follow those of the zeta function. These scenarios can also be imagined by the creation of chaos due to the irregularities in the flow of fluids.

For the case of scattering of HES into two tachyons, the scattering amplitude has the function

$$\mathcal{A} \sim (\sin \alpha)^J \prod_{m=1}^{\infty} [\sin(\pi m \cos^2 \frac{\alpha'}{2})]^{n_m}, \tag{5.10}$$

in terms of $\alpha$, which is an erratic function. The source of this chaos is that the state is the complex superposition of many states with different spin $s$. As mentioned, this source would affect the mutual information and the spreading of quantum information in the hard wall confining model as well. This erratic behavior in MI and $D_c$ in confining models as we have observed in figures 3, 4 ,5 and 6 can also be modeled by such periodic but erratic function as eq. 5.10.

Another point is that the distribution of spacing in the diagram of $D_c$ versus $u_{KK}$ as seen in figure 3, shows log-normal and GUE distribution where the zeros are in the form of $s_n = \frac{1}{2} + iz_n$.

Again, as mentioned in [56], the source of chaotic behavior is the huge degeneracy of string excitations which have the same level $N$ with generic spins.

The first BRST invariant computation that includes the notion of chaos in the string decay was realized in [55]. In [69], using the method of transient chaos analysis, the connections between highly exited strings and a black hole [70–72] has been studied further as the presence of chaos in former is the source of chaos in later. They also depict the *"fractal structure"* or self-similar structure in the plots of scattering angles. These fractal structures also have been observed in the plots of the mutual information in confining backgrounds.

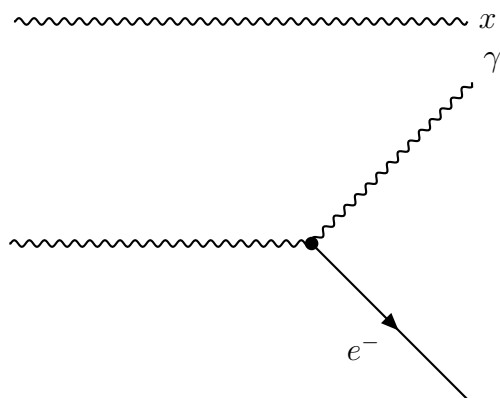

*Figure 17: The scattering process where a photon $\gamma$ scatters off an electron $e^-$ studied in [73]. In the above, $x$ is a witness particle (photon) not participating in the process but is still entangled with the second photon. One can imagine that the two wiggly bosons are the two end points of a an open string, and then calculate the change in the mutual information in this process, as in relation 6.4 here.*

# 6    Entanglement entropy and Compton scattering

In [73], the entanglement entropy of Compton scattering with a witness has been discussed. In their setup, one can imagine that the two entangled photons are the end-points of a string, where one of them is being participated in the Compton scattering and the other one is a witness. The process is shown in figure 17. One can imagine that the two wiggly bosons are actually the two end points of an open string moving on the worldsheet, and then using this picture, and the some similar procedure that has been done for the case of entanglement entropy as in [73], one can calculate the change in the mutual information, in terms of the Stokes parameter, during this process, as we explain next.

First, note that it has been found that the final mutual information of the electron and the witness particle's polarization is non-zero for **low energy** Compton scattering, since due to the scattering process, in spite of no direct interaction between the two particles, they become correlated, which again indicates a direct connection between scattering and mutual information.

The initial pure state can be written as [73, 74]

$$|i\rangle = |e^-, r\rangle (\cos\eta |\gamma, \uparrow\rangle + e^{i\beta}\sin\eta |\gamma, \downarrow\rangle |x, \uparrow\rangle), \tag{6.1}$$

where $\eta$ is the entanglement parameter for the the two photons $\gamma$ and $x$ which can be considered the two end points of a string. Therefore, the parameter $\eta$ is proportional to the string tension $T$. The mass of the particles on the other hand is related to the energy of the string, which can be compared with the position of the hard-wall in confining geometry.

The initial entanglement entropy of $\gamma$ or $x$ can be written in terms of $\gamma$ as [73]

$$S_{EE}^i = -(\cos^2\eta \log\cos^2\eta + \sin^2\eta \log\sin^2\eta). \tag{6.2}$$

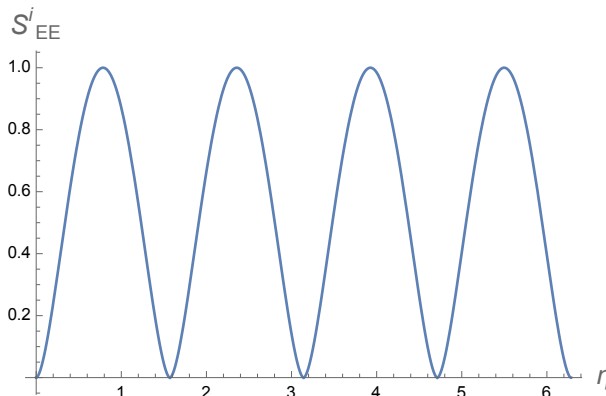

*Figure 18: The relation between the initial entanglement entropy $S_{EE}^i$ between the two particles $\gamma$ and $x$, i.e, relation 6.2, and entanglement parameter $\gamma$. The log are in based 2, so the mutual information would be in bits.*

The relation between $S_{EE}^i$ and $\eta$ is shown in figure 19. One can see a periodic behavior in terms of the entanglement parameter leading to such periodic behavior in mutual information, and then based on the connections between mutual information and string scattering amplitude, the periodic-like patterns we saw in figures 3 and 4, could be expected.

Now one can approximately simulate the process of strings hitting the end wall of the confining geometry with the scattering process of one photon hitting an electron and the other photon acting as a witness, which is actually the other end point of the string. Then, the change in the mutual information during this process, which has been calculated in [73] can be found as

$$\Delta I(e^-, x) = I^f(e^-, x) - I^i(e^-, x) = \Delta S_{EE,e} + \Delta S_{EE,x} - \Delta S_{EE,ex} \geq 0. \tag{6.3}$$

There, the change in the entanglement entropy between the coming electron $e$ and the photon at one end $x$ has been calculated by the relation [73]

$$\Delta S_{EE,ex} = -\frac{1}{2}(1 + 0.67\frac{\sigma_T}{V/T}\frac{\omega^2}{m^2}) \log\left(1 + 0.67\frac{\sigma_T}{V/T}\frac{\omega^2}{m^2}\right)$$
$$-\frac{1}{2}(1 - 0.67\frac{\sigma_T}{V/T}\frac{\omega^2}{m^2}) \log\left(1 - 0.67\frac{\sigma_T}{V/T}\frac{\omega^2}{m^2}\right), \tag{6.4}$$

where $\sigma_T = \frac{8\pi r_e^2}{3}$ is the Thomson scattering cross section, $\omega$ is the initial energy of the photon, $V = (2\pi)^3 \delta^3(0)$ is the un-normalized volume, $m$ is the rest mass of the electron, and $T$ is a parameter in the S-matrix as in $S = 1 + iT$. Note that the Stokes parameter is $\langle \sigma_z \rangle = -\frac{\sigma_T}{V/T}\frac{3\omega}{2m}$, and therefore one can write the above relation and the change in the entanglement entropy and mutual information in terms of the Stokes parameter as well.

In the above process, increasing the ratio between the initial energies of $\gamma$ and the electron, i.e, $(\omega/m)$ would increase the jump in the mutual information during the scattering process. Note

that this rest mass $m$ is directly related to the position of the end-wall in the confining geometry.

One can also be reminded about the the diffraction and Young's double-slit interference experiment where demonstrates the probabilistic nature of quantum mechanics. Recently, it has also been demonstrated that string is a double-slit [75], where the images of strings via Veneziano amplitude have been constructed. The information creating these images could also reveal the entanglement structures of such strings. As the amplitude showed that strings are double slit, the direct connection between Veneziano amplitude and mutual information between the two end-points of strings can be imagined.

Also, in [75], it has been shown that the strings are double-slit, where the slits are the end points of the strings. This model can help to reveal the highly complicated structure of a long string. In confining backgrounds with a hard wall or soft wall, the chaotic behavior of strings become even more complicated and chaotic. However, the similarities between the fractal structure between the scattering amplitude and quantum information measures could reveal further such complex structures, and even the entanglement pattern across fundamental strings.

## 6.1   Two tachyons and one excited string amplitude

In this section, in order to further clarify the relations between string scattering amplitude and spread of quantum information during the process, we consider the example of the decay of a highly excited string into tachyons.

First, if the $m$th mode of a string is being excited $n_m$ times, by $n_m$ photons each with polarization of $\lambda$, where each has momentum $-mq$, ($m$ is an integer and $q$ a null vector), the excited string could be written as [18]

$$\prod_{m=1}^{\infty} (\lambda \cdot A_{-m})^{n_m} |0\rangle, \quad N = \sum_{m=1}^{\infty} m \, n_m. \tag{6.5}$$

Then, in [18], the amplitude of the decay of this excited string into two tachyons could be calculated as

$$\mathcal{A} \propto \prod_{m=1}^{\infty} (p_3 \cdot \zeta P_m(p_3 \cdot q))^{n_m}, \quad p_3 \cdot \zeta = -\sqrt{\frac{N}{2}} \sin \alpha, \quad p_3 \cdot q = -\cos^2 \frac{\alpha}{2}, \quad P_m(a) = \frac{(1+m\,a)_{m-1}}{(m-1)!},$$
$$\tag{6.6}$$

where $\alpha$ is the relative angle between the tachyons' direction and the photons which create the string and $\lambda$ is the polarization of the DDF (Del Giudice, Di Vecchia, Fubini [17]) photons which are transverse to the momentum, i.e, $\lambda \cdot q = 0$. In addition, $(a)_m \equiv \frac{\Gamma(a+m)}{\Gamma(a)} = a(a+1)...(a+m-1)$ is the Pochhammer symbol, unifying the rising and falling factorials.

The main lesson here is that, the more correlated the two systems of $A$ and $B$ be, the smoother the connected bulk wedge they could create and therefore, as the modes of the string $n_m$ become

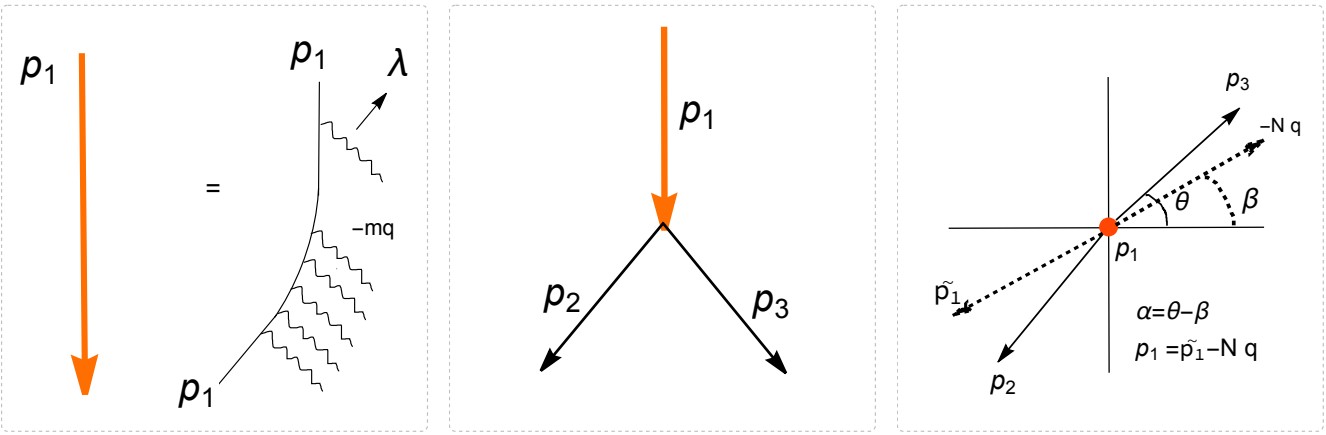

*Figure 19: In the left, the DDF mechanism of exciting a heavy string with $n_m$ photons with momenta $-mq$ off of an initial tachyon with momentum $\tilde{p}_1$ is shown. In the middle, the structure of momentums are shown where the highly excitd string has momentum $p_1$ decaying into tachyons with momenta $p_2$ and $p_3$. In the right part the angles of different momentum vector is shown where the amplitude depends on $\alpha = \theta - \beta$. [18]*

bigger, the scattering amplitude would become bigger as well. This is because in the spacetime building from a more entangled subsystems, the excited strings can have access to photons with the momenta $-mq$ or the null vacuum (vector) $q$ easier. Therefore, one would expect that the mutual information $I$ has a power law relation with $n_m$ as well.

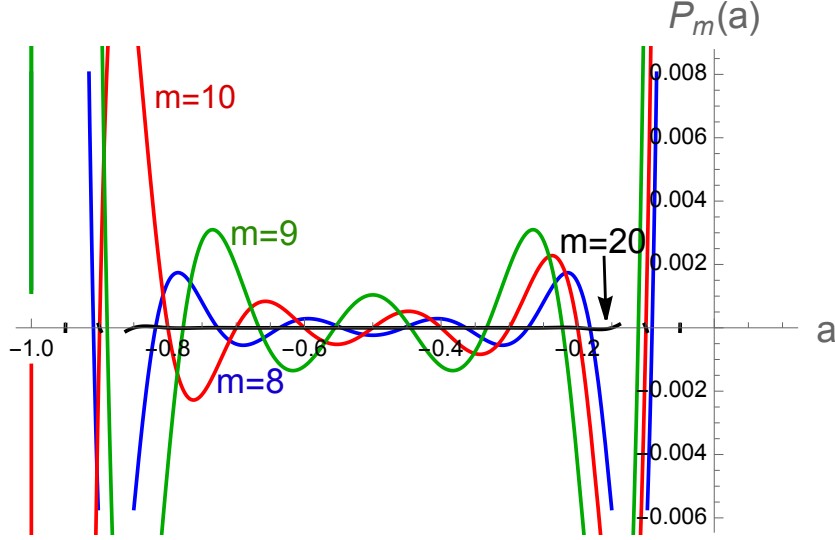

*Figure 20: The function $P_m(a)$ of the Pochhammer symbol $(a)_m$ for various $m$.*

Another interesting fact that this calculation could teach is about the source of the oscillatory behavior in the mutual information. Based on the relation with scattering amplitude, this comes from the Pochhammer function of the Pochhammer symbol $(a)_m$ which its behavior is shown in figure 20. This function is oscillatory for the range of $-1 < a < 0$. By increasing $m$, it becomes more oscillatory but the maximum would be suppressed. The function with bigger $m$ also has more

zeros and also the amplitude is a function of $P_m(a)^{n_m}$ where for bigger $m$ could be estimated by a multiplication of several sine functions leading to a very erratic function, where a small change in the angle $\theta$ causes a large change in the amplitude.

The big $m$ limit corresponds to larger strings, or more excited strings, where more number of modes $n_m$ are being stimulated, which in the confining case corresponds to bigger values of $u_{KK}$ and therefore again one would expect that in larger values of $u_{KK}$, the behavior of mutual information or $D_c$ becomes chaotic, as we have observed in figure 3. So one can propose the following relation for the mutual information $I \propto \prod_{q_b=1}^{q_l} P_{q_b}^{n_{q_b}}$, where $q_b$ is the number of quantum bits that are correlated in the mixed system and $q_l$ is the maximum number of qubits that can be incorporated into the subsystems. So the mutual information is proportional to the Pochhammer function to the power of number of qubits in the system.

Also, in [76], they showed that the positivity of the string scattering amplitude is exactly equivalent to the positivity of entropy, which they expressed as

$$\mathrm{Im}\mathcal{A} > 0 \Longleftrightarrow \mathcal{E}[|\Omega^{\mathrm{prod}}\rangle] \geq 0, \tag{6.7}$$

where $|\Omega^{\mathrm{prod}}\rangle$ is the product of unentangled initial state and $\mathcal{E}[|\Omega\rangle]$ is the linearized entropy, as $\mathcal{E}[|\Omega\rangle] = 1 - \mathrm{Tr}_A[\tilde{\rho}^2]$. Therefore one could imagine that further information from quantum information such as various inequalities in this field could point to further constraints and insights about scattering amplitudes, specifically mutual information and complexity can also point to further details about the mechanism of interactions between strings.

# 7 Kink scattering, chaos and fractal structure

As another example of connections between confinement, chaos and scattering, one can consider the case of kink-antikink and kink-kink scattering process which for the cases of solitons and compactons have been recently studied in [77]. During this process, there is a critical velocity where around it, the behavior would be fractal, and with analogy with black hole physics and AdS/QCD, one would expect that the more radiative the process sets off, the more chaotic it would become.

The periodic relations between the kink-antikink parameters $\lambda_1$ and $\lambda_2$ and their velocities could be caught from the relation

$$u(x,t) = 4 \arctan\left[\left(\frac{\lambda_1 + \lambda_2}{\lambda_1 - \lambda_2}\right) \tan\left(\frac{v_1 - v_2}{4}\right)\right], \tag{7.1}$$

where from the tangent function, the periodic structure could be detected. In addition, the similarities with the zeros of the string scattering amplitude, and also similarities to the patterns of mutual information and critical distance $D_c$ could be noticed here.

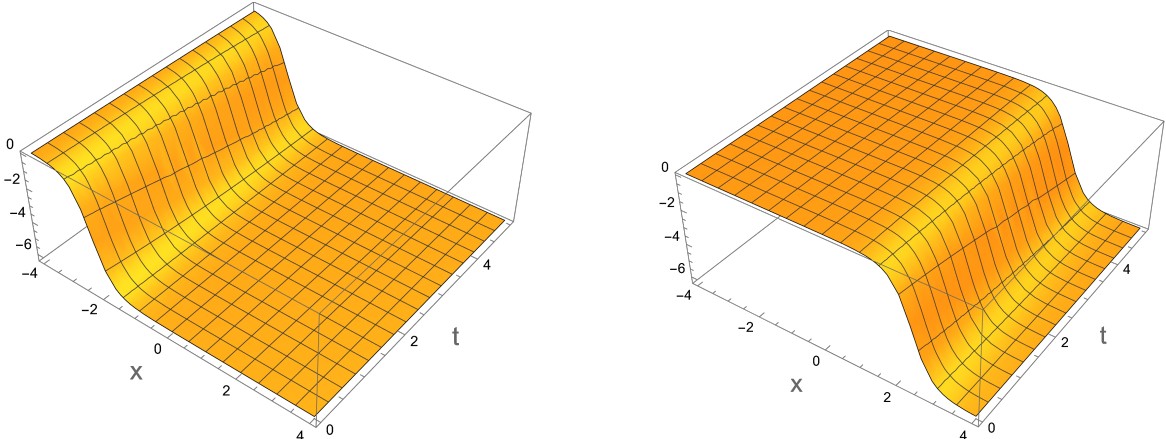

*Figure 21: The kink solution with equation $u(x,t) = -4\arctan(\exp(\lambda x + \frac{t}{\lambda} + \mu))$ and $\lambda > 0$ shown in the left and the anti-kink case with the same relation but with $\lambda < 0$ is shown in the right part. The interaction and scattering of kink-antikink could be compared with the spread of mutual information in confining models, where in both cases the critical phase transitions, and around it the fractal structures could emerge. Specially, the emergence of period behaviors in these fractal-like structures is what we have observed in both cases.*

Periodic structures in the zero's of string scattering amplitude also appear when the highly excited string states involved are non-generic, such as states in the first Regge trajectory. These periodic structures can also be reminiscent of "scar states", related to periodic orbits, in otherwise non-integrable systems, which have been discussed recently by Dodelson and Zhiboedov [78].

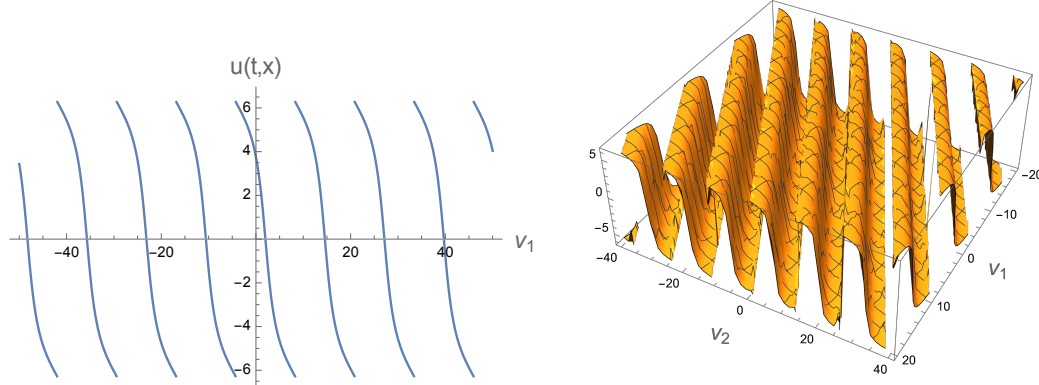

*Figure 22: The periodic structure in the behaviors of kink and anti-kink velocities, which correlates with the periodic structures in the behavior of mutual information and $D_c$ and also string scattering amplitude.*

Note that this critical velocity where around it the fractal behavior emerges, corresponds to the specific $u_{KK}$ where the plot of $D_c$ or mutual information shows the fractal behavior as well. This can be seen in our figure 4, as the fractal structures emerge around $u_{KK} \simeq 0.39$.

An important point is that the spectrum of radiation would scatter several times and as time passes and by bouncing back several times, it makes the fractals which has been discussed in [79] for the case of scattering of oscillation. The quantum information would behave similar to the

radiation as well and therefore one would expect such bouncing and creation of fractals in the confining geometries, or in the backgrounds with solitons or compactons, as we have demonstrated here for the case of mutual information. Note that for the case of spreading of quantum information and mutual information, similar to the case of radiation, one would expect the formation of "compact shockwaves" as well. One should particularly compare figure 9 of [77] with our figures for $D_c$ where it shows many similar fractal-like behavior. So the exchange of, and the distribution of entanglement entropy and mutual information, could give a lot of information about the interactions of topological defects as well.

As an example of scattering in a topological kink solution, one first could consider a Heaviside-structured potential like the one considered in [77] as

$$V(\eta) = \sum_{n=-\infty}^{\infty} \left( |\eta - 4n| - \frac{1}{2}(\eta - 4n)^2 \right) H_n(\eta),$$
$$H_n(\eta) := \theta(\eta - 4n + 2)\theta(\eta - 4n - 2), \tag{7.2}$$

where its behavior and its derivative are shown in figure 23. The confining solitonic Skyrme model can be considered as an example of having such potential. Here, $\eta$ is a real scalar field in $1 + 1$ dimension with the action

$$S = \int dt dx \left[ \frac{1}{2}(\partial_t \eta)^2 - \frac{1}{2}(\partial_x \eta)^2 - V(\eta) \right], \tag{7.3}$$

and with the BPS equation of $\frac{d\eta}{dx} = \pm\sqrt{2V(\eta)}$.

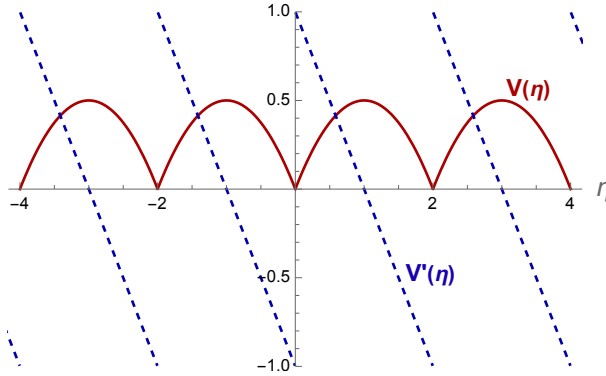

Figure 23: The potential of a topological kink solution or a solitonic Skyrme model.

In such interactions between compact topological defects, where the spread of information in confining geometries with a wall at the end of the geometry could be regarded as such, two main classification classes for the scattering could exist, namely the kink-antikink pairs which oscillate and radiate, and the kink-antikink pairs which themselves emerge from the collisions where a shockwave is being formed and this shockwave would decay into a "cascade of compact oscillons" [77]. These are in-fact the zeros that [37, 56] observed where their specific intervals

demonstrate chaos in the whole of the scattering process.

Another point is that higher scattering velocities would cause smaller shockwaves and radiation and the other way around. The particular oscillating behavior could only be observed for $v < v_c$. That is why in the massive cases where the graviton has a finite mass which causes the scattering velocities become suppressed, more noises could be noticed in the structures of mutual information, entanglement entropy, complexity and complexity of purification as well [33]. The critical velocity $v_c$ would depend on the parameters of the model, such as mass of graviton $m$, charge of the particles $q$, dimension of the background $d$, temperature $T$, position of the hard wall or soft wall $u_{KK}$ in confining geometries, the fluxes, etc. This critical velocity is the initial velocity of kinks where a single bounce scattering would change to multi-bounce or annihilation which then the "collective coordinate model" could be applied.

Additionally, the escape and no-escape cases, which as mentioned in [77], alternate in a fractal manner close to the critical velocity $v_c$, correspond to fractal structures around $D_c$, where the mutual information between the subsystems drop to zero, creating different behaviors for the entangled pairs and structures of mutual information in various phases. The "**collective coordinate model**" could be implemented in describing all these collisions and also the structures of quantum information and entanglement.

Also, note that during phase transition, one would expect the appearance of compactons and other topological kinks in the mechanical systems [80], which again as we have studied in the first part of the work, mutual information and critical distance $D_c$ could be a measure for detecting such compactons, specially in confining models. Note that as these fields are not differentiable measures, the dispersion relation $\omega(k)$ could not be particularly used well to picture the evolution and structures of the data and information, and therefore complexity, complexity of purification and mutual information as measures of computational work and strength of perturbation of the initial state would act as better measures.

# 8 Chaos, Regge conformal block, pole-skipping and quantum error correction

Another interesting relation between scattering properties and quantum information measures can come from studying the Regge trajectories. Regge theory is a framework in quantum field theory that describes high energy scattering amplitudes. In the context of conformal field theory, Regge conformal blocks are certain correlation functions that appear in the operator product expansion of CFT correlation functions. The Regge limit is associated with the high energy behavior of the scattering amplitudes, and conformal blocks could capture the exchange of conformal primary operators in the OPE. Exploring the interplay between Regge theory and conformal blocks would then reveal insights into the behavior of the scattering amplitudes in CFT.

On the other hand, Pole-skipping is a phenomenon observed in certain scattering amplitudes where the behavior of poles in the complex angular momentum plane differs from what is expected in a standard Regge trajectory. The study of pole-skipping phenomena then involve analyzing the structures of amplitudes and the behavior of singularities. Understanding how pole-skipping manifests in the context of CFT and conformal blocks could provide insights into the analytic properties of correlation functions which is the aim of this section.

Quantum error correction codes are techniques used in quantum information theory to protect quantum information from errors. The connection between bulk reconstruction and quantum error correction codes actually would involve understanding how information in the bulk theory is encoded and protected in the dual CFT. Quantum error correction code has been suggested as a mechanism of bulk emergence from boundary CFT data and for the bulk reconstruction [81]. Its connections with other models of bulk reconstruction such as modular flow has been studied in [32].

The idea of this section is to discuss the links between Regge conformal blocks, pole-skipping phenomena and bulk reconstruction from the quantum error correction codes. Note that the relationship between Regge conformal blocks and pole-skipping have already been discussed in [82]. In [82], it has been shown that the analysis of the symmetric and traceless fields with any integer spin in the near-horizon of Rindler-AdS black hole can capture the Regge behavior of conformal blocks. We can show that this connection would be another basis for the quantum-error correcting behavior of the bulk [81,83].

First note that the chaos bound

$$\lambda_L \leq \frac{2\pi}{\beta},\tag{8.1}$$

is actually related to the Regge growth bound $j \leq 2$,

$$\left|\mathcal{M}(s, t \leq 0)\right| \propto s^j \qquad (s \to \infty),\tag{8.2}$$

which assumed the AdS/CFT correspondence in the flat spacetime limit as AdS length $R \to \infty$. This bound would be related to the bound on the eigenvalues of modular Hamiltonian [38], i.e, modular chaos bound.

Another important quantity is the Mellin transform which is an integral transform and is closely related to the Laplace and Fourier transforms, and for a function $f$ is defined as

$$[\mathcal{M}f](s) = \varphi(s) = \int_0^\infty x^{s-1} f(x) dx,\tag{8.3}$$

and the inverse transform is

$$[\mathcal{M}^{-1}\varphi](x) = f(x) = \frac{1}{2\pi i}\int_{c-i\infty}^{c+i\infty} x^{-s}\varphi(s)ds, \tag{8.4}$$

where it is a line integral taken over a vertical line in the complex plane and its real part $c$ would meet certain conditions.

These two correspond to quantum channel and exact quantum recovery channels respectively. So an approximate version of the Mellin transform should exist which corresponds to the approximate quantum error correction codes and approximate quantum recovery channels.

For instance the Petz map as a quantum recovery channel can be written as

$$\mathcal{P}_{B\rightarrow A}^{\sigma,\mathcal{N}}(\omega_B) := \sigma_A^{1/2}\mathcal{N}^{\dagger}(\mathcal{N}(\sigma_A)^{-1/2}\omega_B\mathcal{N}(\sigma_A)^{-1/2})\sigma_A^{1/2}, \tag{8.5}$$

where $\sigma_A$ is the quantum state, $\omega_B$ is the input density and $\mathcal{N}_{A\rightarrow B}$ is the quantum channel which takes the system from $A$ to $B$.

Also, the twirled Petz map could be written as

$$\mathcal{R}_{\sigma,\mathcal{N}} := \int_R dt\beta_0(t)\sigma^{-\frac{it}{2}}\mathcal{P}_{\sigma,\mathcal{N}}[\mathcal{N}[\sigma]^{\frac{it}{2}}(.)\mathcal{N}[\sigma]^{-\frac{it}{2}}]\sigma^{\frac{it}{2}}, \tag{8.6}$$

where $\mathcal{P}_{\sigma,\mathcal{N}}$ is the normal Petz map, $\beta_0$ is the probability density in the form $\beta_0(t) := \frac{\pi}{2}(\cosh(\pi t) + 1)^{-1}$, and $\mathcal{N}^{\star}$ is the adjoint of channel $\mathcal{N}$.

The Mellin transform could be connected to quantum recovery channel as will be discussed. Note that the in [84,85] it has been pointed out that a natural language for AdS/CFT would be in Mellin space and holographic S-matrix. Since quantum recovery channel and also modular Hamiltonian are two tools for the bulk reconstruction, we expect that they would also have connections with the S-matrix and Mellin amplitude.

As the Mellin transform would be important in probability theory, this could point out to its connection to quantum error correction codes. In probability theory, the Mellin transform is used for studying the distribution of products of random variables. For a random variable $X$ with its positive part $X^+ = \max[X, 0]$ and negative part $X^- = \max[-X, 0]$, the Mellin transform of $X$ is

$$\mathcal{M}_X(s) = \int_0^{\infty} x^s dF_{X^+}(x) + \gamma\int_0^{\infty} x^s dF_{X^-}(x), \tag{8.7}$$

where $\gamma$ is a formal indeterminate with $\gamma^2 = 1$, and for two independent random variables $X$ and $Y$, the Mellin transform of their product would be

$$\mathcal{M}_{XY}(s) = \mathcal{M}_X(s)\mathcal{M}_Y(s). \tag{8.8}$$

Note that the connected scalar four-point function in CFT could be written as

$$\langle \mathcal{O}_1(x_1)\mathcal{O}_2(x_2)\mathcal{O}(x_3)\mathcal{O}(x_4)\rangle_c = \int d\delta_{ij} M(\delta_{ij}) \prod_{1\leq i<j\leq 4} \Gamma(\delta_{ij})(x_{ij}^2)^{-\delta_{ij}}, \tag{8.9}$$

where $\sum_{j=1}^4 \delta_{ij} = 0$ and $-\delta_{ii} = \Delta_i$.

The Mellin formula of the bulk S-matrix $\mathcal{M}(s_{ij})$ could be written as

$$\mathcal{M}(s_{ij}) = \frac{1}{\mathcal{N}} \lim_{R\to\infty} \int_{-i\infty}^{+i\infty} \frac{d\alpha}{2\pi i} e^\alpha \alpha^{\frac{d-\Sigma\Delta_i}{2}} M(\delta_{ij} = -\frac{R^2}{4\alpha} s_{ij}), \tag{8.10}$$

where $\mathcal{M}(s_{ij})$ is the bulk S-matrix in the flat spacetime limit, $s_{ij}$ are the Mandelstam variables, $R$ is the AdS length, and $M(\delta_{ij})$ is the Mellin amplitude of large $N$ CFT.

In [86], a new relation for connection between the boundary operator $\mathcal{O}$ and the bulk operator $\phi_a$ which has support in the entanglement wedge $a$ has been found as

$$\mathcal{O} := -\frac{1}{d_{\text{code}}} \int_R dt \beta_0(t) e^{\frac{1}{2}(1-it)H_A} \text{Tr}_{\bar{A}}[J(\phi_a \otimes 1_{\bar{a}})J^\dagger] e^{\frac{1}{2}(1+it)H_A}, \tag{8.11}$$

where $H_A = -\log(J\tau J^\dagger)_A$ is the modular Hamiltonian in the boundary for the subregion $A$ which is associated with the maximally mixed state $\tau$ on the code subspace or one could write it as the logarithmic directional derivative as

$$\mathcal{O} := \mathcal{R}^\star[\phi_a] = -\frac{1}{d_{\text{code}}} \frac{d}{dt}\Big|_{t=0} H_A[\tau_{\text{code}} + t\phi_a \otimes 1_{\bar{a}}]. \tag{8.12}$$

Since both the Mellin transform formalism and quantum error correction could lead to AdS/CFT and bulk reconstruction, we expect to derive connections between them as well.

This bulk S-matrix in the flat limit would be related to the out-of-time-order correlator (OTOC), of large $N$ CFT in Rindler spacetime written as

$$A(u,v) \approx \int_{-i\infty}^{+i\infty} \frac{dT}{4i} \sigma^T \Gamma\left(\frac{2\Delta_1 - T}{2}\right) \Gamma\left(\frac{2\Delta_3 - T}{2}\right) e^{-i\pi T/2} \int_\infty^{+\infty} dx M(ix, T)(x/2)^{T-2} e^{ix\sigma\cosh\rho}, \tag{8.13}$$

via the Mellin amplitude $M$.

Now, for different limits, these quantities could be evaluated and their behaviors with the behavior of modular Hamiltonian and quantum error correction could be compared to find the similarities. For instance, for large time, $t \gg 1$, the OTOC would behave as $A(u,v) \propto \sigma^{1-j} \approx e^{(j-1)t}$ which is related to the large $s$, i.e, $s \to \infty$, where $\mathcal{M}(s_{ij}) \propto s^j$. Considering other limits and areas in parameter space can make this connection precise further.

# 9 Conclusion

In this work, first, we showed in holographic confining backgrounds, and at low Mandelstam variable $s$, there are connections between the logarithmic branch cut singularities of the open string scattering amplitude and the behaviors of the peaks or singular behaviors observed in the mutual information and critical distance $D_c$. These peaks actually come from the dependence of the string tension on the holographic radial coordinate and are a sign of the creation of bound states at low energies. In addition, one can imagine that the two entangled subregions spread quantum information across the spacetime in the modular time and the wavelets interfere with each other, and also after hitting the wall, they bounce back and create chaos structures in the form we detected the signs from the figures of $D_c$ and the string scattering amplitude.

The power low decay of both quantities match at larger energies, $s$ or $u_{KK}$, where the real part of both follows a $s^{-1}$ curve. This observation can further establish the links between the patterns of entanglement entropy and the string scattering amplitude, and also can further strengthen the ER=EPR conjecture. In addition, it proposes that the string scattering amplitude can serve in detecting the phase transitions. Hence, one could expect that in confining backgrounds, the phase diagrams of various mixed quantum correlation measures such as mutual information or critical distance $D_c$, negativity, entanglement and complexity of purification at one side, and the open string scattering amplitude, $\mathcal{A}$, on the other side would be connected.

We also discussed the behavior of scatterings of classical pinballs, scattering of strings in a background with a black hole, and scatterings of highly excited strings, all in the presence of two correlated mixed systems with a connected wedge, and then discussed similarities and differences for each case. In this regard, we used modular flows and modular Hamiltonians, as the foundation for introducing modular time for the behavior of entanglement patterns, so to draw connections between scattering amplitude, binding energy and the spread of mutual quantum information in these setups. Additionally, we discussed how mutual information can detect chaos. The leaky torus structures have also been used to model the spread of quantum information and the emergence of chaos in different confining backgrounds which has an end-wall in their IR affecting the chaotic structures.

To get more detailed formulations, the exchange of entanglement entropy, and the change in the mutual information for the Compton scattering of two photons, and also the decay of a highly excited string into two tachyons have been discussed, where the source of many periodic structures could be examined further. The scattering of kink-antikinks, and the emergence of chaos in this scenario have also been examined, where the links with the previous sections have been drawn. Finally, in our final section, the relationships between Regge conformal block and quantum error correction codes, through pole-skipping, chaos bound, and their behaviors in various limits have been illustrated.

## acknowledgments

This work has been supported by an appointment to the JRG Program at the APCTP through the Science and Technology Promotion Fund and Lottery Fund of the Korean Government. It has also been supported by the Korean Local Governments - Gyeongsangbuk-do Province and Pohang City - and by the National Research Foundation of Korea (NRF) funded by the Korean government (MSIT) (grant numbers 2021R1A2C1010834).

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
