# Peer review of "String amplitudes and mutual information in confining backgrounds: the partonic behavior"

_SciPost Physics_

## Round 3 · Referee Report · Maurizio Firrotta · 2024-4-6

Report

The manuscript is valid, well written and well motivated.

Before recommending for publication, I would suggest some modifications

*)Page 1, line 6 from below:
the first computation of covariant scattering amplitudes in the DDF formalism was not in reference [16] but in the following reference :
Nucl.Phys.B 952 (2020) 114943
I suggest to include the reference and also to be more precise about the literature.

*)Page 16, Chaos in string amplitudes was also recently studied in more general processes:
JHEP 04 (2023) 052
2312.02127
2401.02211
I suggest to include the references.

*)page 18, line 6 below formula (5.6) there is a missprint:
ransom matrix -> random matrix

*)page 21, line 8 from above:
in reference [52] the authors did not find any fractal structure in the string amplitudes, but they had some argument about it.
I would like to stress that chaos does not imply automatically fractal structure.

In reference [54] there is no any definition of chaos in the string decay. The first BRST invariant computation that includes the notion of chaos in the string decay was realized in
JHEP 04 (2023) 052.
I suggest to include the reference and also to be more precise about the literature.

---

## Round 3 · Referee Report · Anonymous · 2024-4-7

Report

In this manuscript, the author studied the connection between string amplitudes and mutual information in holographic confining backgrounds. The main finding, as the author summarized in the introduction section, is a certain analogy between (1) the dependence of the critical distance D_c as a function of the IR scale u_{KK} of the confining theory, and (2) the scattering amplitude as a function of the Mandelstam variable s. To help clarify this analogy, the author further drew connections to a several other phenomena, including modular flow, quantum error correction codes, etc. Despite the intriguing claims and the possibility of tying several fields together, I’m not convinced by the analogy the author is trying to propose.

For one, the figures 2, 3, 4 and 5 seem very suspicious. To elaborate, the physical quantity being considered is the mutual information between two spatial slabs separated by distance D, in a confining theory with (inverse) mass scale u_{KK}. As is well known, as one increases the distance D, there is a phase transition happening at D_c where the RT surface of the two slabs become disconnected, and therefore the mutual information vanishes in the large N limit. One would naturally expect, since the metric depends on u_{KK} in a simple and analytic way, the quantity D_c would be an (at lease piecewise) analytic function of u_{KK}. However, figure 2 shows the dependence is highly non-analytic and fluctuating. I can only find the same behavior in a previous paper by the same author and no clear mathematic explanation for the erratic behavior is given. On the other hand, the way the lines are broken in figures 2, 3, 4 and 5 appear to be a typical feature of instability in numerical integrations. With no quantitative equations and physical interpretation to back up, these plots appear suspicious and significantly weaken the findings that are based on it.

Nonetheless, let’s suppose the dependence of D_c on u_{KK} is indeed erratic and has the features as the plots are showing, the connection to the scattering amplitude still appears to be vague. The connections proposed include the branch cut structures as well as the asymptotic fall-off of the two functions. Neither of these were demonstrated with clear quantitative analysis in the manuscript, but rather established by inspecting the plots. We should also note that the quantities that are being compared, D_c versus amplitude A, u_{KK} versus Mandelstam s, have different dimensions, so a direct comparison does not appear meaningful without clear physical justification. In my opinion, the manuscript is not successful in doing so.

Therefore, in my opinion, the main finding in the manuscript is shaky. Apart from this, the overall writing of the manuscript seems to have the tendency of overextending analogies rather than trying to present solid derivations. Based on these, I do not believe it matches the criterion in order to be published on SciPost.

---

## Round 5 · List of Changes

I addressed all the critics of my first referee. Explained further why mutual information, string amplitude, and chaos in confining geometries should be related using the mathematical relations for the connections between entanglement entropy and scattering amplitude.
I also explained how modular Hamiltonian should come into play. Also add a few figures to explain my setup better to show how the end-wall of confining geometry causes the appearance of chaos.
Also, I added a few references.

---

## Editorial Decision

unknown